# SAVVY: Spatial Awareness via Audio-Visual LLMs through Seeing and Hearing

**Mingfei Chen** [*†]  **Zijun Cui** [*†]  **Xiulong Liu** [*†]

**Jinlin Xiang** [†]  **Yang Zheng** [†]  **Jingyuan Li** [†]  **Eli Shlizerman** [‡§]

## Abstract

3D spatial reasoning in dynamic, audio-visual environments is a cornerstone of human cognition yet remains largely unexplored by existing Audio-Visual Large Language Models (AV-LLMs) and benchmarks, which predominantly focus on static or 2D scenes. We introduce SAVVY-Bench, the first benchmark for 3D spatial reasoning in dynamic scenes with synchronized spatial audio. SAVVY-Bench is comprised of thousands of carefully curated question–answer pairs probing both directional and distance relationships involving static and moving objects, and requires fine-grained temporal grounding, consistent 3D localization, and multi-modal annotation. To tackle this challenge, we propose SAVVY, a novel training-free reasoning pipeline that consists of two stages: (i) Egocentric Spatial Tracks Estimation, which leverages AV-LLMs as well as other audio-visual methods to track the trajectories of key objects related to the query using both visual and spatial audio cues, and (ii) Dynamic Global Map Construction, which aggregates multi-modal queried object trajectories and converts them into a unified global dynamic map. Using the constructed map, a final QA answer is obtained through a coordinate transformation that aligns the global map with the queried viewpoint. Empirical evaluation demonstrates that SAVVY substantially enhances performance of state-of-the-art AV-LLMs, setting a new standard and stage for approaching dynamic 3D spatial reasoning in AV-LLMs.

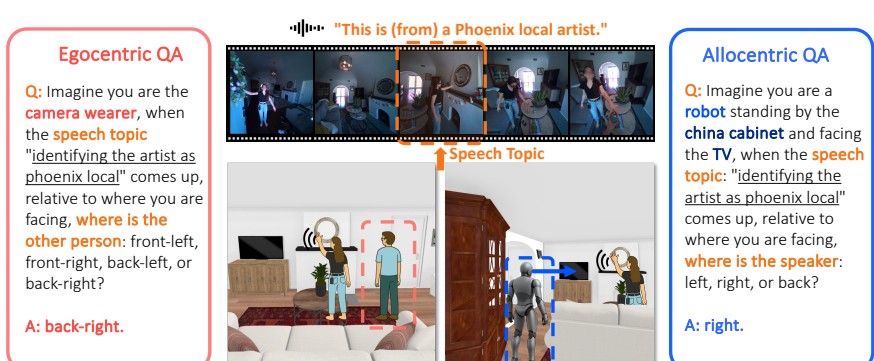

Figure 1: 3D spatial reasoning in dynamic audio-visual environments. The task requires fine-grained 3D question answering across egocentric and allocentric frames in dynamic scenes.

---

[*]These authors contributed equally.

[†]Department of Electrical & Computer Engineering, University of Washington, Seattle, USA.

[‡]Department of Applied Mathematics, University of Washington, Seattle, USA

[§]Corresponding author: shlizee@uw.edu

39th Conference on Neural Information Processing Systems (NeurIPS 2025).

# 1 Introduction

3D spatial reasoning in dynamic scenes is a core aspect of human intelligence, allowing us to navigate and understand changing environments. Imagine watching an egocentric video where a person wearing a head-mounted camera guides someone through a multi-room apartment. During such navigation, a question could arise as shown in "Allocentric QA" denoted in Figure 1. To answer such a question, a human would engage in several mental processes: (i) identify the moments when the referenced speech event occurs and locate the relevant objects ("*China Cabinet*", "*TV*", "*Speaker*") in space; (ii) convert egocentric observations into an allocentric map anchored at the *China Cabinet* and oriented towards the *TV*; and (iii) mentally compute the *Speaker*'s position within this allocentric map. While humans perform these steps naturally, they are cognitively demanding, especially under dynamic, shifting viewpoints, as demonstrated in early human cognitive studies [1]. This raises a key question: can existing foundation models such as Multi-Modal LLMs (MLLMs) reason about dynamic 3D scenes with spatial intelligence?

Despite growing interest in grounding foundation models in 3D environments, most existing works remain limited to static scenes. Indeed, previous spatial reasoning benchmarks [2, 3] targeted static visual environments with no moving objects. However, real-world scenarios are usually dynamic and involve diverse moving objects and sounds. Existing foundation models that support spatial reasoning in 3D such as [4, 2, 5] assume a static world, and thus cannot generalize to dynamic scenarios. Moreover, they rely exclusively on visual input, neglecting the critical role of spatial audio in capturing semantics and spatial cues beyond the visual field. These limitations highlight the need for benchmarks and models capable of dynamic 3D spatial reasoning across both audio and visual modalities. We refer to such models as Audio-Visual LLMs (AV-LLMs), which are MLLMs that jointly reason over audio and visual inputs.

To fill these gaps, we introduce SAVVY-Bench, a first-of-its-kind benchmark designed for 3D spatial reasoning in dynamic scenes for AV-LLMs. A key feature of SAVVY-Bench is its coverage of both egocentric and allocentric question types: some questions require reasoning from the camera wearer's viewpoint (egocentric), while others rely on fixed external references (allocentric), as depicted in Figure 1. SAVVY-Bench comprises thousands of QA pairs that probe spatial relationships involving both static and dynamic objects, focusing on distance and directional aspects. In terms of modalities, SAVVY-Bench targets audio-visual question answering with a strong emphasis on moving objects. To support fine-grained spatial reasoning, we incorporate multi-channel audio that captures directional information beyond what is visible in the video.

Beyond benchmark construction, enabling effective spatial reasoning in 3D dynamic scenes remains a challenge. We impose that an effective AV-LLM for reasoning for such environments must (i) achieve robust temporal grounding to locate keyframes and detect relevant objects, (ii) develop spatial perception in both visual and auditory (spatial audio) to track locations of objects from egocentric views, and (iii) transform egocentric observations into a consistent global coordinate frame to accurately reason about spatial relationships. Existing video-language models unable to fully incorporate spatial reasoning and egocentric-allocentric perspective transformation even in static visual scenes [3], while dynamic 3D environments add further complexity, which require tracking the state of moving objects. Moreover, existing AV-LLMs typically rely on monaural audio input. This reliance limits access to spatial audio cues and restricts the model's ability to support human-like spatial understanding.

To support these capabilities, we introduce SAVVY, a training-free pipeline that augments AV-LLMs with structured spatial reasoning, integrating spatial audio cues and egocentric-to-global mapping. It operates in two stages: (i) Extracting sparse "snapshot" descriptions of key events and objects via an AV-LLM, and constructing egocentric tracks by estimating object direction and distance relative to the camera from video and spatial audio; these tracks align auditory and visual signals at key timestamps relevant to the query. (ii) Aggregating these tracks into a dynamic global map for accurate reasoning over both egocentric and allocentric queries. We perform experiments with the proposed pipeline on SAVVY-Bench. Extensive experiments demonstrate that SAVVY performs best in comparison to existing state-of-the-art AV-LLMs.

To summarize our contributions: **(i)** We introduce SAVVY-Bench, the first spatial reasoning benchmark for dynamic 3D scenes, with an integration of both (spatial) audio and visual modality. **(ii)** We propose a training-free pipeline that augments existing AV-LLMs with strong spatial reasoning capabilities. **(iii)** Experiments on SAVVY-Bench show that significantly outperforms existing AV-

| Dataset | Modality | Dynamic Scene | Cross-Room Spatial QA | Allocentric | Direction | Distance |
|---|---|---|---|---|---|---|
| EgoSchema [6] | V | ✗ | ✗ | ✗ | ✗ | ✗ |
| OpenEQA [7] | V | ✗ | ✗ | ✗ | ✓ | ✗ |
| MUSIC-AVQA [8] | A+V | ✗ | ✗ | ✗ | ✓ | ✗ |
| VSI-Bench [3] | V | ✗ | ✓ | ✓ | ✓ | ✓ |
| Ego4D-AVD [9] | A+V | ✓ | ✗ | ✗ | ✓ | ✗ |
| **SAVVY-Bench (Ours)** | A+V | ✓ | ✓ | ✓ | ✓ | ✓ |

Table 1: **Comparison of SAVVY-Bench with other Visual and Audio-Visual Benchmarks.** SAVVY-Bench focuses on spatial relations (distance and direction) among objects to evaluate 3D spatial reasoning in large and dynamic audio-visual scenes.

LLMs on dynamic spatial QA task, with a significant improvement of **+7.1%** on overall QA accuracy against even the best performing AV-LLMs (Gemini-2.5 Pro).

## 2 Related Works

### 2.1 Multi-modal Large Language Models for Spatial Reasoning

Recent advances in Multi-modal Large Language Models (MLLMs) have extended language models to process visual [10, 11, 12, 13, 14, 15, 16] and audio [17, 18, 19, 20] modalities, giving rise to Audio-Visual LLMs (AV-LLMs) [21, 22, 23, 24, 25, 26, 27, 28]. However, most MLLMs and AV-LLMs remain limited in spatial reasoning capabilities. While some models incorporate basic 2D localization [29, 30, 31], spatial reasoning remains largely unaddressed due to reliance on 2D training data and the lack of large-scale 3D annotations. Recent efforts incorporate 3D information via point clouds [2, 32], or spatial scene representations such as graphs [4, 33, 34], voxel grids [35, 5, 36, 37], maps [38, 3], and neural fields [39, 40]. However, these models are limited to static environments and do not support dynamic scenes.

Moreover, visual cues alone are insufficient for spatial reasoning. In dynamic scenes, where objects leave the visual field, spatial audio provides critical cues for localization which is missing in existing AV-LLMs [41] since these downmix multi-channel audio to mono and discard spatial information. Extracting spatial cues from audio remains challenging due to the complexity of real-world soundscapes and the lack of high-quality, localized annotations. While learning-based spatial audio localization methods [42, 43, 44] exist, they are trained on synthetic data with specific receiver configurations, and do not generalize to real-world environments that are typically noisy and reverberant.

### 2.2 Benchmarks for Multi-Modal Understanding and Reasoning

Existing benchmarks for evaluating MLLMs primarily focus on semantic understanding from images [45, 46, 47] or video inputs [48, 49, 50, 51, 6]. For image inputs, benchmarks such as MMBench [47], MMMU [46], and MM-Vet [45] assess reasoning across diverse domains but do not address temporal aspects. Video-based benchmarks like MVBench [49], EgoTaskQA [51], EgoSchema [6] and additional works [52, 53, 54, 23] focus on event-based or temporal concept understanding in either exocentric or egocentric views, while they do not address 3D spatial relationships in dynamic scenes. When the modality extends to both visual and audio, benchmarks such as MUSIC-AVQA [8, 55], Ego4D AV Diarization [9] and other [56, 57, 58, 22] address sounding events as well as spatial relationships between sounding objects in 2D image plane, without addressing 3D relations. Benchmarks such as ScanQA [59] and OpenEQA [7] introduce spatial reasoning in 3D environments. These focus solely on static layouts and coarse spatial relations. The closest benchmark to SAVVY-Bench is the VSI-Bench [3], which leverages 3D information for fine-grained visual spatial reasoning, but is constrained to static scenes only. In contrast, **SAVVY-Bench** is the first benchmark for *audio-visual* spatial reasoning in *dynamic* scenes. Table 1 illustrates a detailed comparison with related benchmarks.

## 3 SAVVY-Bench

### 3.1 Overview

SAVVY-Bench is the first benchmark for evaluation of 3D spatial reasoning of AV-LLMs in dynamic, multi-room scenes. It builds on a manually selected subset of the Aria Everyday Activities

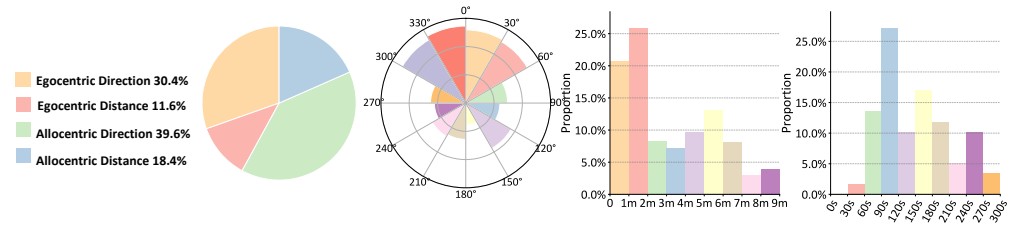

(a) QA Task Distribution   (b) Angle Distribution   (c) Distance Distribution   (d) Video Duration Distribution

Figure 2: **Benchmark Statistics.** (a) Task distribution by type. (b) Angle distribution of egocentric queries over 360°. (c) Distribution of egocentric query distances. (d) Video duration distribution.

Dataset(AEA), Meta Reality Labs-R [60], which includes over 600 sound events across 58 daily-life scenarios. Each scenario provides the synchronized visual input and the spatial audio captured by 7-microphone array on Aria glasses. SAVVY-Bench poses queries spatial relations among moving and static entities in 3D space.

**Task Taxonomy.** SAVVY-Bench defines 4 spatial-relational QA tasks across two reference frames: **egocentric** (camera-centered) and **allocentric** (object-centered) (examples shown in Figure 1). Each question is anchored to a sound event and requires reasoning regarding the relative direction and the absolute distance between a sounding object and a reference point. In egocentric tasks, the reference is the camera wearer; In allocentric tasks, it is a hypothetical robot positioned net to a single static object and faces another object. Directional reasoning is posed as a multiple-choice question, offering 3 options (left, right, back) for simpler layouts and 4 options (front-left, front-right, back-left, back-right) for more complex ones. Distance reasoning requires to provide a numeric estimate of the distance in meters.

**Statistics.** Figure 2(a) shows the distribution of QA tasks: Egocentric Direction (30.4%), Egocentric Distance (11.6%), Allocentric Distance (18.4%), and Allocentric Direction (39.6%). Relative direction questions cover the full 360° azimuth (Figure 2(b)), including the challenging cases that involve rear angles (90–270°)in Egocentric QA, where the target sounding object is not within the camera view. Distance values range from <0.5 to 9 meters (Figure 2(c)). Video durations span from 30 to 300s (Figure 2(d)).

### 3.2 Benchmark Construction

We develop a systematic data pipeline to generate high-quality question–answer pairs for SAVVY-Bench. The pipeline includes four stages: **Data Preprocessing**, **Annotation**, **QA Synthesis**, and **Quality Review**. In **Data Preprocessing**, fisheye videos from the AEA dataset are undistorted to a rectilinear format for compatibility with AV-LLM inputs. Multiview videos are temporally aligned into a unified timeline, and audio is extracted into seven-channel wav file. In **Annotation**, we utilize proprietary AV-LLMs [41] to extract word-level transcriptions, speech topics, and sound events. Object locations are detected in 3D using EFM3D [61] and manually refined in a point-cloud interface. Human trajectories are extracted from aligned camera data, recovering both the location and the orientation for all speakers. All annotations are manually calibrated to align spatial and event data. In **QA Synthesis**, structured QA pairs are generated using templates applied to the annotated metadata. The **Quality Review** stage involves human verification to ensure that each QA pair is clear, grounded, and unambiguous. Further details are provided in Supplementary Materials.

## 4 SAVVY

### 4.1 Formulation and Overview

Given a video with $N_C$ spatial audio channels and a natural language question $\mathcal{Q}$, the goal is to predict the relative direction or absolute distance of a dynamic *target object* (i.e., a sounding object) during an audio event. Each question is framed from either an **egocentric** (camera-centered) or **allocentric** (object-centered) perspective (Section 3.1). To bridge multimodal input and spatial reasoning, we

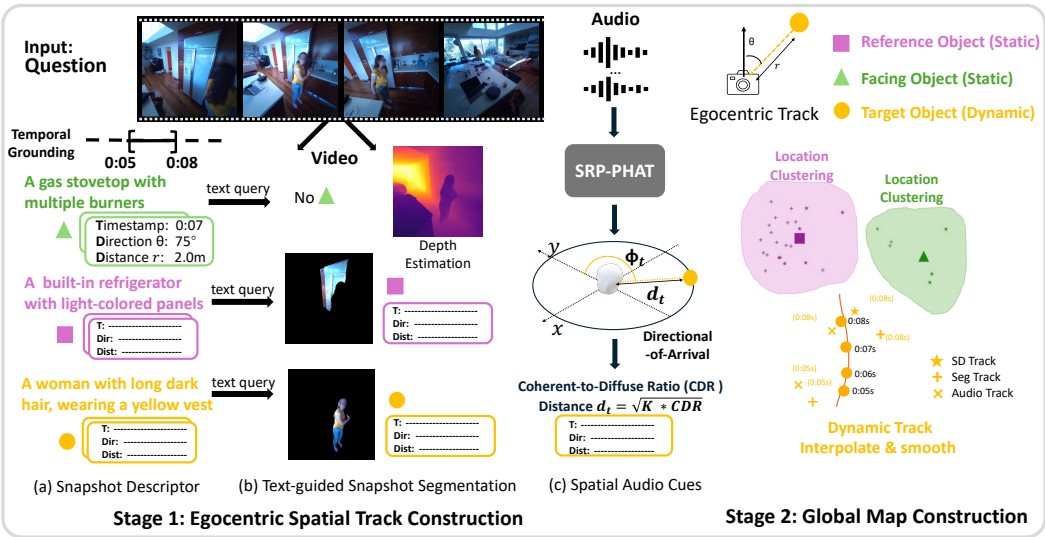

Figure 3: **SAVVY** consists of two stages: Given a query and video with spatial audio, stage 1 extracts Egocentric Spatial Tracks with (a) "Snapshot" Descriptors via AV-LLMs, (b) Text-Guided Snapshot Segmentation, and (c) Spatial Audio Cues. Stage 2 constructs a dynamic Global Map by converting egocentric tracks to global coordinates, clustering static objects, and smoothing dynamic trajectories.

introduce **SAVVY**, a training-free plugin pipeline that augments AV-LLMs by extracting structured spatial information from visual, audio, and language inputs in two stages (Figure 3):

**Stage 1: Egocentric Spatial Track Construction.** We estimate a per-frame egocentric trajectory for each object referenced by $\mathcal{Q}$, using cues from vision, language, and spatial audio. Each trajectory is defined as $\{(t, \theta, r)\}$, where $t$ is the timestamp, $\theta \in [-180°, 180°]$ is the azimuth (0° front, -90° left, 90° right), and $r$ is the distance in meters from the camera location.

**Stage 2: Dynamic Global Map Construction.** Egocentric tracks are projected onto a 2D $xy$-plane using the SLAM-derived [62] camera trajectory $\mathbf{L}(t) \in \mathbb{R}^2$: $\mathbf{p}(t) = \mathbf{L}(t) + \begin{bmatrix} r \cdot \cos(\theta) \\ r \cdot \sin(\theta) \end{bmatrix}$.

The target forms a global trajectory $\{\mathbf{p}_{\text{sound}}(t) \mid t \in \mathcal{T}_q\}$, while the reference and facing objects are treated as static, with global positions $\mathbf{p}_{\text{ref}}$ and $\mathbf{p}_{\text{face}}$ computed by averaging their tracks. These define the **dynamic global map**: $\mathcal{M}_q = \{\mathbf{p}_{\text{sound}}(t) \mid t \in \mathcal{T}_q\} \cup \{\mathbf{p}_{\text{ref}}, \mathbf{p}_{\text{face}}\}$. To answer $\mathcal{Q}$, SAVVY uses $\mathcal{M}_q$ to compute the direction and the distance of the target relative to the camera (egocentric) or in an object-centered frame.

## 4.2 Stage 1: Egocentric Spatial Tracks Estimation

We estimate egocentric spatial tracks with three components (Figure 3(a), (b) and (c)):

**Snapshot Descriptor.** Given $\mathcal{Q}$ and video, we prompt AV-LLM once to generate structured *snapshot description*. The model first determines the relevant time span $\mathcal{T}_q$ (temporal grounding) corresponding to the query-referenced event, as well as whether the question is framed egocentrically or allocentrically. It then identifies up to three object roles: the *target* object (sound source), the *reference* object (anchor for allocentric frame), and the *facing* object (defines orientation). Out of the three roles, egocentric queries require the target object only, while allocentric queries require all three to define a third-party coordinate frame. Each object is represented by a descriptive textual phrase and an egocentric trajectory, given as a sequence of $(t, \theta, r)$ tuples—timestamp, direction, and distance (Figure 3(a)).

**Text-Guided Snapshot Segmentation.** Snapshot descriptors provide sparse spatial cues and often omit intermediate frames, resulting in incomplete trajectories — particularly for dynamic objects or static objects that are visible briefly. To address this, we use visual foundation models to recover missing egocentric trajectory segments (Figure 3(b)). We uniformly sample $N$ frames from the video and use the textual descriptions generated by the Snapshot Descriptor module as queries for

text-guided segmentation. For each sampled frame, we segment the *target*, *reference*, or *facing* object using foundation models such as CLIPSeg [63] and SAM2 [64], following prior work [2]. From each object mask, we compute the centroid relative to the image center to estimate the azimuth angle $\theta$ with respect to the camera orientation. In parallel, we apply a monocular metric depth estimator [65] to predict the object distance $r$. This yields an egocentric trajectory of up to $N$ points per object.

**Spatial Audio Cues.** Spatial audio provides spatial cues that complement visual input for robust tracking. We estimate both the direction and the distance of sound sources using multi-channel audio recorded by wearable microphone arrays. The method supports training-free, geometry-aware tracking in complex acoustic environments. Specifically, to estimate direction-of-arrival (DoA), we adopt the SRP-PHAT algorithm [66]. Let $M$ microphones at positions $\mathbf{p}_m = (x_m, y_m, z_m)^\top$ record audio sampled at $f_s$ Hz, we compute time-domain cross-correlation $R_{mn}$ for each microphone pair $(m, n)$. For each candidate azimuth $\phi$ with unit direction vector $\mathbf{u}(\phi) = [\cos\phi, \sin\phi, 0]^\top$, the inter-channel delay $\tau_{mn}(\phi) = \frac{(\mathbf{p}_m - \mathbf{p}_n)^\top \mathbf{u}(\phi)}{c}$ is quantized into an integer lag $\ell_{mn}(\phi) = \mathrm{round}(\tau_{mn}(\phi) f_s)$. DoA is estimated by maximizing the steered response power: $\hat{\phi} = \arg\max_\phi P(\phi)$, where $P(\phi) = \sum_{m=1}^{M-1} \sum_{n=m+1}^{M} R_{mn}[\ell_{mn}(\phi)]$.

To estimate distance, we adopt the coherent-to-diffuse ratio (CDR) approach [67]. We compute CDR at each time frame and use distance estimates from the visual-guided modules (Snapshot Descriptor and text-guided snapshot segmentation) to exploit the acoustic property that $D_t^2 \cdot \mathrm{CDR}_t$ remains approximately constant in a given environment. We estimate this constant $K$ by computing $D_t^2 \cdot \mathrm{CDR}_t$ per frame $t$, applying DBSCAN [68] to filter outliers, and minimizing the squared error over remaining frames: $\hat{d}_t = \sqrt{\frac{K}{\mathrm{CDR}_t}}$, where $K = \arg\min_K \sum_t \left( D_t^2 \cdot \widehat{\mathrm{CDR}}_t - K \right)^2$.

To reduce interference from the camera wearer's voice or front-facing background noise, we discard detections within a narrow $[-5°, 5°]$ range around the forward axis. This filtering improves reliability in egocentric direction estimation. Together, these estimates of the direction and the distance yield per-frame egocentric trajectories and serve as spatial audio cues.

### 4.3 Stage 2: Dynamic Global Map Construction

For reasoning of spatial relationships, SAVVY aggregates the three egocentric trajectories from Stage 1 into a unified global map. Each per-frame track is transformed to global coordinates, resulting in a 2D spatial map representation suitable for downstream spatial reasoning.

The track aggregation process is illustrated in Figure 3(c). For static objects (e.g., *reference* or *facing*), globalized positions are clustered using DBSCAN to suppress outliers, and the centroid of the dominant cluster is used as the final location. For dynamic *target* objects, a time-varying trajectory $\mathbf{p}(t)$ is constructed by filtering temporally aligned outputs from the three egocentric tracks in Stage 1 and mapping them to global coordinates. A Kalman filter [69] is applied to interpolate and smooth $\mathbf{p}(t)$, producing a continuous and robust path.

The final map $\mathcal{M}_q$ contains a continuous trajectory for the *target* object and static positions for the *reference* and *facing* objects. SAVVY then resolves the target's location based on the predicted query type from the Snapshot Descriptor: either egocentric (relative to the camera) or allocentric. In the allocentric case, the reference-to-facing vector is aligned with the positive $y$-axis, and the map is rotated accordingly before computing the target's relative position.

## 5 Experiments

### 5.1 Metrics

**SAVVY-Bench.** SAVVY-Bench includes questions with respect to relative direction and absolute distance for both egocentric and allocentric categories (Section 3.1). Direction questions (*dir*) are multiple choice, and we report **accuracy** based on exact or fuzzy matching [3]. For distance (*dist*), which ranges from less than 1 m to more than 8 m, we avoid target-scaling [3]. Instead, we compute the **average relative accuracy** across absolute error thresholds from 0.1 m to 1.0 m (step size 0.1 m) to allow fair comparison across varying distances.

| Method | Egocentric | | Allocentric | | |
| | Dir | Dist | Dir | Dist | overall |
|---|---|---|---|---|---|
| Chance-Level (Freq) | 30.2 | - | 32.1 | - | - |
| Human-Level | 93.5 | 71.2 | 94.0 | 56.3 | 78.7 |
| LongVALE [22] | 41.9 | 8.3 | 26.9 | 18.3 | 23.8 |
| video-SALMONN [26] | 35.4 | 36.9 | 24.9 | 14.5 | 27.9 |
| Ola [24] | 43.0 | 32.5 | 25.2 | 26.2 | 31.7 |
| VideoLLaMA2-7B [21] | 46.4 | 42.7 | 24.8 | 15.2 | 32.3 |
| MiniCPM-o 2.6 [25] | 45.8 | 42.3 | 25.2 | 15.9 | 32.3 |
| EgoGPT [23] | 40.2 | 57.6 | 25.4 | 22.8 | 36.5 |
| Gemini-2.5-flash | 74.2 | 49.7 | 29.8 | 29.0 | 45.7 |
| Gemini-2.5-pro | 75.2 | 59.6 | 31.7 | 37.0 | 50.9 |
| SAVVY | **84.7** | **62.9** | **44.0** | **40.2** | **58.0** |

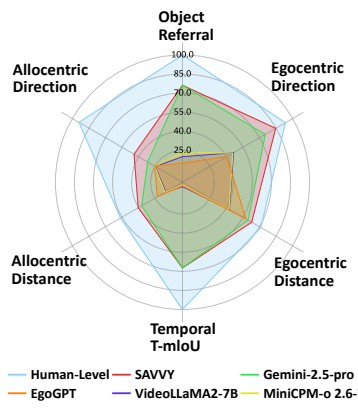

Table 2: Evaluation on SAVVY-Bench. Left: Accuracy on egocentric and allocentric QAs. Right: Radar plot showing QA and SD-Eval accuracy comparison for top-3 open-source AV-LLMs.

**Snapshot Descriptor Evaluation.** To better understand the capabilities of AV-LLMs on SAVVY-Bench, we evaluate two tasks aligned with the Snapshot Descriptor (Section 4.2), reported under "SD Eval" in Table 3. (i) *Temporal grounding task* measures how accurately a model localizes the queried sound event in time. We use Intersection over Union (IoU) [70] between the predicted and groundtruth time intervals. Performance is reported as Recall@1, averaged over IoU thresholds from 0.05 to 0.5 (step size 0.05), and summarized as mean IoU (**t-mIoU**). (ii) *Object referral task* tests whether the model correctly describes objects given the question and video. Egocentric questions involve only the *target* sounding object; allocentric questions require further identification of the *reference* and the *facing* objects. We compute accuracy for (**referral**) via string matching and LLM-based judging [41], with all required objects expected to match.

**Localization Accuracy.** We assess localization by comparing predicted and groundtruth positions. We propose a new metric, **localization accuracy** (*loc_acc*, in Tables 4, 5 and 6): a predicted location is correct if the direction angular error $\theta\_err$ is below $45°$ and the distance error $r\_err$ is below 1 m.

## 5.2  Main Results

**Benchmark Models.** We evaluate 8 AV-LLMs as listed in Table 2. Out of these, 6 are open models designed for joint audio and video understanding. These include models that add an audio branch to a video-language model: VideoLLaMA2 [21], LongVALE [22], and EgoGPT [23], with EgoGPT fine-tuned on egocentric data. Video-SALMONN [26] adds a visual encoder to an audio-language model. Ola [24] and MiniCPM-o-2.6 [25] are trained as omni-modal models. Most models have around 7B parameters; MiniCPM-o-2.6 has 8B and Video-SALMONN has 13B. We also evaluate 2 proprietary AV-LLMs: Gemini-2.5-pro and Gemini-2.5-flash. All open-source AV-LLMs are evaluated using 32 sampled video frames and mono-channel compressed audio input. We include one chance-level baseline based on *Frequency*—for multiple-choice direction tasks, and the human-level baseline by aggregating independent responses of 6 annotators. For prompts, inference settings of all AV-LLMs, and human evaluation guidelines, please refer to the supplementary materials.

**Human-level Performance.** Humans achieve 78.7% accuracy on SAVVY-Bench, outperforming SAVVY (ours), the best method, by 20.7%. Directional tasks yield near-perfect human performance (93.5–94.0%), reaffirming strong intuitive spatial reasoning. In distance estimation and in egocentric settings humans score 71.2% compared to 62.9% for the best model. For allocentric distance, human accuracy drops to 56.3%, reflecting the added difficulty of measuring distance after coordinate transformation involving various *reference / facing* objects.

**AV-LLMs Results.** All AV-LLMs perform better on egocentric QA than on allocentric QA. Most models perform at or below chance on the allocentric relative direction task. In contrast, performance on the egocentric version is higher; Gemini-2.5 models reach up to 75% accuracy. For absolute distance estimation, proprietary models outperform open-source ones on egocentric tasks. Some open-source models, e.g. Ego-GPT, show accuracy gaps up to 34.8% between egocentric and allocentric

| Method | referral | t-mIoU |
|---|---|---|
| LongVALE [22] | 33.7 | 0.7 |
| VideoLLaMA2-7B [21] | 20.0 | 3.5 |
| MiniCPM-o 2.6 [25] | 23.2 | 2.3 |
| EgoGPT [23] | 14.9 | 2.8 |
| Ola [24] | 21.2 | 3.0 |
| Gemini-2.5-flash | 66.2 | 42.6 |
| Gemini-2.5-pro | **76.2** | **67.4** |

Table 3: SD-Eval accuracy on temporal grounding (*t-mIoU*) and object *referral*.

| Mic | Localization | | | DoA | |
|---|---|---|---|---|---|
| | loc_acc↑ | $\theta$_err↓ | $r$_err↓ | l/r↑ | f/b↑ |
| 02 | 19.5 | 104.8° | 1.86m | 76.2 | **55.8** |
| 34 | 18.6 | 112.1° | **1.33m** | **82.3** | 54.5 |
| 56 | 23.6 | **100.7°** | 2.11m | 81.6 | 55.5 |
| 0234 | 15.5 | 116.4° | 1.54m | 78.4 | 52.6 |
| 0256 | 44.2 | **39.2°** | 1.25m | 79.8 | 69.8 |
| 3456 | **44.3** | 45.6° | **1.11m** | 81.8 | **75.0** |

Table 4: Sounding object localization and Direction of Arrival (DoA) accuracy on left/right (l/r) and front/back (f/b) across different microphones.

| Type | loc_acc↑ | $\theta$_err↓ | $r$_err↓ |
|---|---|---|---|
| *Target Sounding Object:* | | | |
| SD | 50.0 | 43.0° | **0.84m** |
| Seg | **72.4** | **25.6°** | 0.85m |
| Audio | 44.3 | 45.6° | 1.11m |
| *Reference/Facing Object:* | | | |
| SD | 33.8 | 58.6° | **1.10m** |
| Seg | **38.3** | **57.7°** | 1.29m |

Table 5: Object localization results of various egocentric track types.

| Track Type | | | Sound | Egocentric | | Allocentric | |
|---|---|---|---|---|---|---|---|
| SD | Audio | Seg | loc_acc | dir | dist | dir | dist |
| ✓ | | | 55.7 | 68.3 | 47.9 | 42.4 | 38.9 |
| | ✓ | | 59.0 | 73.9 | 48.1 | - | - |
| | | ✓ | 72.5 | 81.2 | 52.0 | 34.2 | 23.1 |
| ✓ | ✓ | | 66.8 | 74.5 | 54.6 | **44.4** | **41.0** |
| | ✓ | ✓ | 73.6 | 84.2 | 57.7 | 34.4 | 24.0 |
| ✓ | | ✓ | 76.0 | 83.2 | 54.4 | 44.2 | 36.9 |
| ✓ | ✓ | ✓ | **78.6** | **84.7** | **62.9** | 44.0 | 40.2 |

Table 6: Egocentric track aggregation ablations on sounding object localization and SAVVY-Bench QA.

distance tasks. Compared with egocentric questions, allocentric ones require more complex spatial transformation and reasoning of static *reference/ facing* objects that appear briefly in the video.

To better assess AV-LLM capabilities on SAVVY-Bench, we evaluate 2 additional tasks, temporal grounding and object referral, as detailed in *SD-Eval* (5.1). Table 3 shows most open-source models achieve under 5% temporal mIoU, indicating poor event-time alignment. Synchronizing complex events like speech remains challenging for models at 7B-parameter scale. In object referral, fewer than 35% of responses are correctly grounded. These limitations may stem from AV-LLM training objectives that prioritize caption-level alignment and visual grounding, rather than learning to synchronize event timelines and spatial object tracks across audio and visual streams, a key requirement for complex spatial reasoning. Gemini-2.5-pro improves referral accuracy by 10.0% and temporal grounding by 24.8% over Gemini-2.5-flash, suggesting the benefits of more advanced temporal-spatial reasoning capabilities.

**SAVVY.** As shown in Table 2, adding SAVVY as a plugin to Gemini-2.5-pro—without additional training or multi-turn AV-LLM inference—substantially improves relative direction accuracy: +9.5% for egocentric and +12.3% for allocentric QA. Distance accuracy also improves in both settings. SAVVY integrates Snapshot Descriptor, text-guided snapshot segmentation, spatial audio cues, and explicit spatial transformations to ground reasoning in a global map. These components collectively demonstrate a modular path toward enhancing spatial reasoning for AV-LLMs, and motivate future work on training LLMs to internalize such structured spatial reasoning abilities.

### 5.3 Ablations and Analysis

We analyze how each egocentric track component—Snapshot Descriptor (*SD*), text-guided snapshot segmentation (*Seg*), and spatial audio-based tracks (*Audio*)—contributes to spatial reasoning individually and comprehensively.

**Egocentric Tracks.** Table 5 reports object localization performance for each egocentric track type (Stage 1 before aggregation), using the metrics from Section 5.1. Given a single image, *Seg* achieves the highest localization accuracy (*loc_acc*) and the lowest relative angle error ($\theta$_err) for all objects type. For distance estimation, *SD* yields the lowest distance error ($d\_err$), benefiting from temporal context and the advanced reasoning capabilities of AV-LLMs, as illustrated in the allocentric distance example in Figure 4. *Audio* tracks perform competitively for sounding object localization, comparable to *SD*, using spatial audio cues alone. Table 4 studies the impact of different microphone channel combinations (more details in the supplementary materials) on localization and

DoA accuracy. Microphone combinations that include both front and rear positions (e.g., 0256, 3456) significantly improve front/back (*f/b*) DoA accuracy, while all configurations yield strong left/right (*l/r*) performance due to symmetric mic placement. Our SAVVY uses setup 3456, which achieves 81.8% (*l/r*), 75.0% (*f/b*) on DoA, and the highest localization accuracy (44.3%).

**Track Aggregation for Global Map Construction.** Table 6 examines how different combinations of egocentric track sources (*SD*, *Audio*, and *Seg*) in Stage 2 of global map construction affect spatial QA accuracy and sounding object localization (*loc_acc*) at the query moment. **SD** alone yields strong performance on allocentric QA, improving direction accuracy by 10.7% over Gemini-2.5-pro due to more precise static object localization. However, without audio, *SD* underperforms on egocentric QA due to limited tracking capability for dynamic sounding objects. **Audio-only** tracking achieves comparable egocentric direction accuracy to Gemini-2.5-pro. **SD+Audio** combines the strengths of both components, improving localization accuracy by +7.8% over Audio-only and boosting allocentric direction and distance QA by +12.7% and +4.0% over Gemini-2.5-pro respectively. **Seg**-based tracks achieve the highest standalone localization accuracy (72.5%). It also achieves very high egocentric direction accuracy due to precise localization of sound sources. However, it shows clear weaknesses in allocentric QA compared to *SD*, likely due to weaker static reference grounding from uniformly sampled frames. In contrast, *SD* leverages Gemini's stronger reasoning ability to more reliably identify static *reference / facing* objects. Finally, **SAVVY** integrates all track types, achieving the highest localization accuracy (78.6%) and the best egocentric QA performance—outperforming *Seg* by +3.5% on direction and *SD+Audio* by +8.3% on distance, its closest competitors in each task. A slight drop in allocentric QA is observed relative to *SD+Audio* due to noise introduced by *Seg*'s static estimates, but overall SAVVY delivers the most balanced and accurate spatial reasoning across tasks.

These findings highlight the complementary strengths of each modality: 1) *Seg* significantly contributes to localizing dynamic sounding objects, improving direction grounding, especially for egocentric questions. 2) *Audio* cues enhance distance estimation largely and also improve both localization and direction QA accuracy. 3) *SD* performs best on allocentric reasoning by more accurately localizing reference and facing objects to construct meaningful spatial relationships. These results underscore the value of integrated reasoning across modalities for robust spatial understanding.

## 5.4 Qualitative Analysis

Figure 4 presents two example cases from Gemini-2.5-pro, the strongest AV-LLM model in our experiments: the top shows egocentric direction reasoning, and the bottom shows allocentric distance reasoning. These illustrate the model's step-by-step reasoning and how SAVVY addresses its errors.

**How do AV-LLMs perform spatial reasoning using video and monoaural audio?** AV-LLMs process video with compressed monoaural audio, while SAVVY-Bench tasks require spatial sound localization—a task humans perform via binaural hearing. In the egocentric direction task (Figure 4, top), Gemini-2.5-pro links sound to visible objects—in this case, identifying "the other person" as the sound source—grounds the event time (0:39–0:41), tracks the sound source and camera wearer's trajectory, and infers direction. For distance measurement (Figure 4, bottom), the model further relies on visual cues and commonsense priors.

**What errors do AV-LLMs make in spatial reasoning?** Common errors of AV-LLMs on SAVVY-Bench often root in temporal grounding, object referral, and spatial relations (direction and distance). While Table 3 reports accuracy on the first two, the examples in Figure 4 highlight spatial relation errors. In the queried event, the sound source ("the other person") disappears from view for tens of seconds. Gemini-2.5-pro infers its trajectory based only on its last visible location, leading to incorrect sound source location estimation. Because the model underutilizes spatial audio—despite its key role in human egocentric perception—it performs modestly when the object appears briefly but fails when it is absent for longer durations.

**How does SAVVY address these errors?** In Figure 4, SAVVY uses spatial audio cues to correctly localize the sound source at approximately 130° (back-right). The *SD* and *Seg* modules lack egocentric tracks in the visual context, but audio enables correct tracking of the dynamic sound source, yielding accurate back-right direction inference. In the allocentric distance case, *SD* and *Seg* help localize the egocentric track of the reference object—a two-seater coffee table—reasonably well. Combined with the accurate track of the sounding object, SAVVY produces a correct distance estimate. By

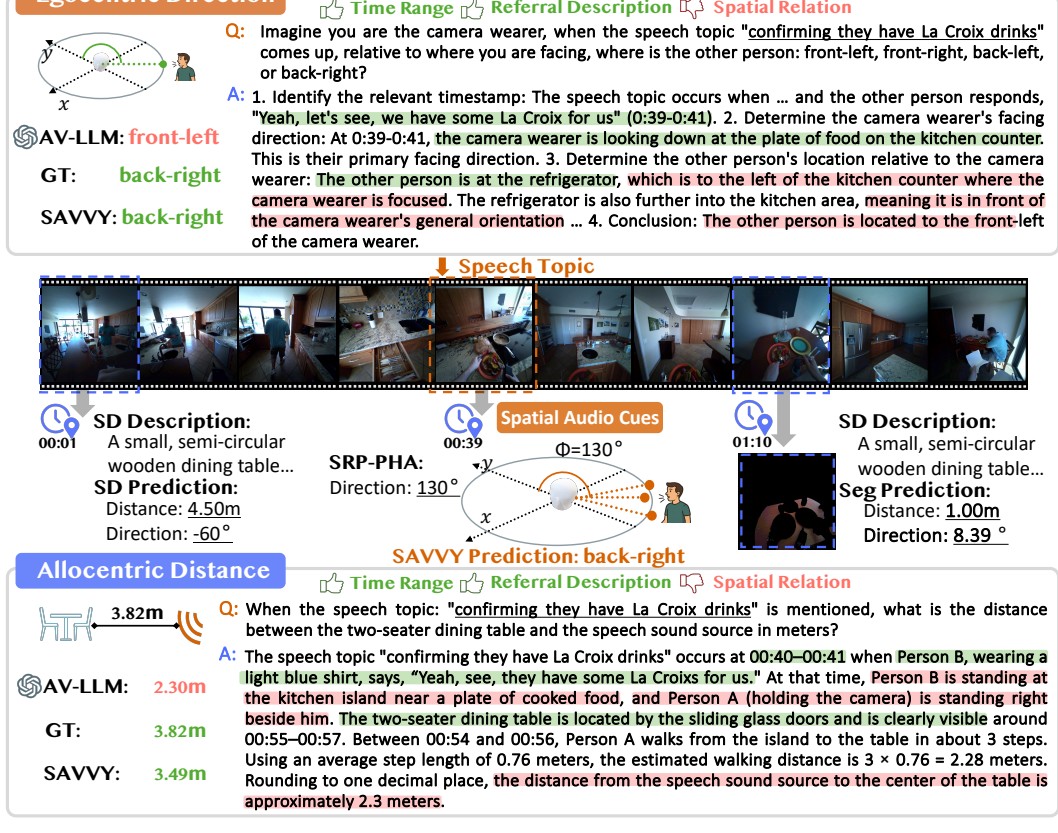

Figure 4: **Example reasoning process of AV-LLMs. Top (Egocentric direction); Bottom (Allocentric distance). Middle (SAVVY):** SAVVY successfully fixes the spatial relation errors.

combining snapshot descriptors, segmentation, spatial audio cues, and explicit coordinate mapping, SAVVY offers a proof of concept for potential solutions of improving AV-LLM spatial reasoning.

## 6 Conclusions

We introduce SAVVY-Bench and SAVVY, the first benchmark and training-free pipeline for 3D spatial reasoning in dynamic audio–visual environments. SAVVY-Bench poses thousands of questions grounded in egocentric video and multi-channel audio, covering both egocentric and allocentric perspectives. It targets core spatial skills—direction, distance, temporal grounding, and grounded object referral with focused evaluations on these aspects. Building on this, SAVVY boosts spatial reasoning over standard AV-LLMs by integrating snapshot-based perception, audio–visual tracking, and dynamic global mapping. Together, they offer a practical foundation for advancing spatial intelligence in multimodal AI systems.

Looking ahead, SAVVY is designed to be modular and extensible. While current data is based on indoor Aria captures, the spatial audio–video pipeline naturally generalizes to outdoor and driving scenarios. As richer spatial computing devices emerge, SAVVY-Bench can connect high-quality data, models, and applications, helping catalyze a robust ecosystem for spatial AI.

## 7 Acknowledgments

The authors acknowledge the partial support of HDR Institute: Accelerated AI Algorithms for Data-Driven Discovery (A3D3) National Science Foundation grant PHY-2117997. The authors also acknowledge the partial support by the Departments of Electrical Computer Engineering and Applied Mathematics. The authors are thankful to the eScience Center at the University of Washington.

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

# A Summary of Supplementary Materials

In this supplementary materials, we provide:

1. A video demonstrating the case examples detailed in Figure 4 of the main paper is available at our webpage here. For the best viewing experience, **we recommend watching the video with headphone or a device that supports spatial audio playback.** See Section B for details.

2. Details of benchmark construction pipeline, including data processing, annotations, QA synthesis and quality review, see Section C.

3. Evaluation details of SAVVY-Bench, including open-source AV-LLMs, proprietary AV-LLMs and human evaluations, see Section D.

4. Details of input data to the pipeline, including video input settings, multi-channel audio settings (microphone configurations), as well as camera trajectory, see Section E.

5. Additional implementation details of all stages in SAVVY, see Section F.

6. Efficiency analysis of SAVVY, see Section G.

7. Additional ablation studies of SAVVY-Bench on input modalities, see Blind Testing in Section H.

8. Limitations of SAVVY, see Section I.

9. Broader impacts of the work with safeguards, see Section J.

10. Additional qualitative results which showcase the reasoning process of SAVVY as well as the error types analysis, see Section K.

# B Video Examples

The demo videos contain two case examples—one egocentric direction task and one allocentric distance task—captured in a single video clip featuring two people conversing in an indoor setting. **We recommend watching the video with headphones or a device that supports spatial audio playback.**

These examples correspond to the qualitative results presented in the main paper. In both cases, the queried event is: *confirming they have La Croix drinks*, corresponding to the spoken sentence, "Yeah, let's see ... grab some La Croix for us," from a guest (a male wearing a blue shirt) speaking to the camera wearer.

**Egocentric Direction Example.** The question asks for the relative direction of the other person, with options: *front-left, front-right, back-left*, or *back-right*. In this clip, the other person is not visible at any timestamp during the event, as he is located in the *back-right* quadrant relative to the camera wearer. While the direction must be inferred from spatial audio cues, a human viewer can clearly perceive the sound as coming from the back-right when watching the video with spatial audio. SAVVY correctly predicts this as *back-right*, whereas Gemini-2.5-pro incorrectly classifies it as *front-left*.

**Allocentric Distance Example.** This question asks for the distance between the two-seater dining table and the speech sound source (the male guest in the blue shirt). The table is clearly visible in several frames throughout the video. SAVVY localizes both the table and the sound source using a combination of egocentric tracks via Snapshot Descriptor, text-guided snapshot segmentation and spatial audio cues. SAVVY estimates the distance as *3.49 meters*, which is close to the ground truth of *3.82 meters*. In contrast, Gemini-2.5-pro predicts a significantly incorrect distance of *2.30 meters*.

These examples illustrate SAVVY's robustness in both directional and quantitative spatial reasoning, especially in challenging, partially observed scenarios.

# C Benchmark Construction

We implement a four-stage pipeline to construct SAVVY-Bench. The stages are **Data Preprocessing**, **Annotation**, **QA Synthesis**, and **Quality Review**. Each stage combines automated tools with human checks to ensure that every Question–Answer (QA) pair is precise.

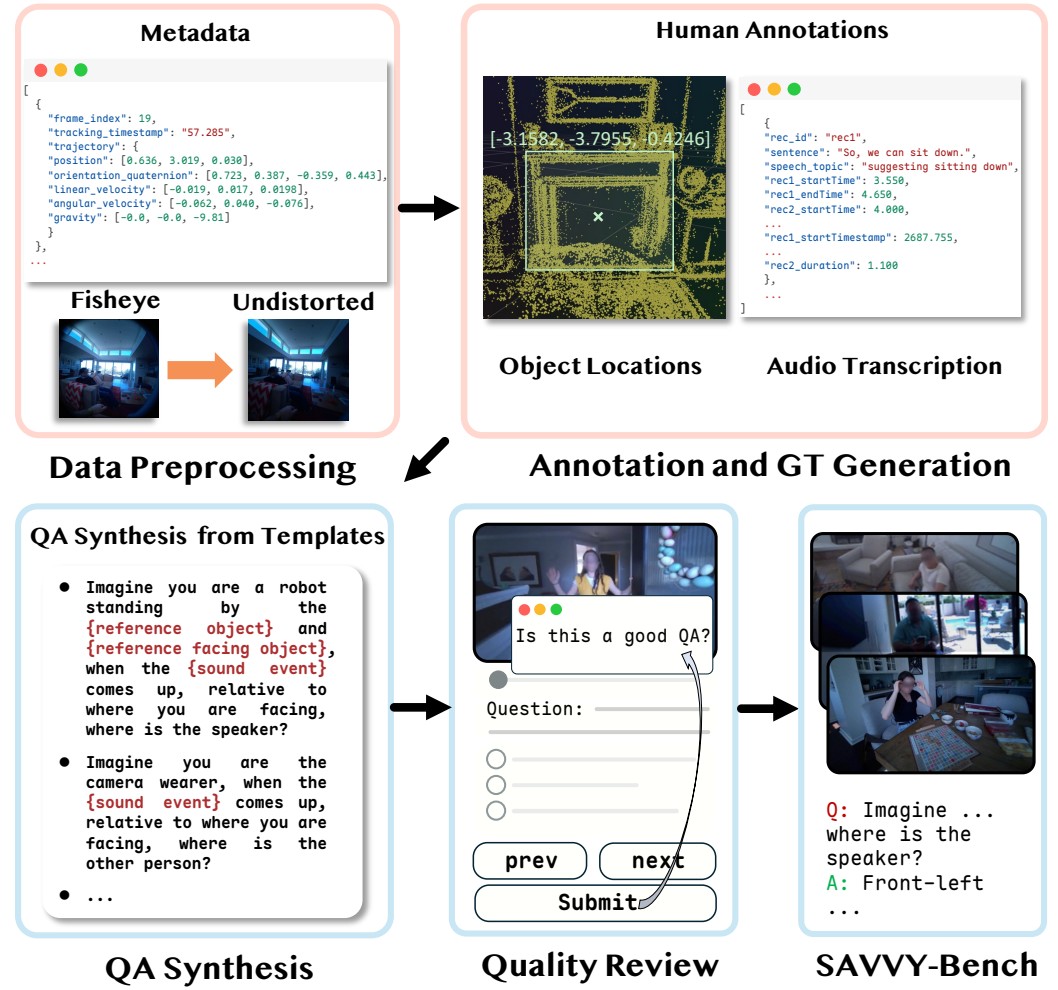

Figure 5: Human-in-the-Loop Dataset Curation and Benchmark Construction Workflow for SAVVY-Bench.

## C.1 Data Preprocessing

We preprocess the video data from the Aria Everyday Activities (AEA) Dataset [60] and integrate raw annotations—such as word-level transcriptions, camera-wearer trajectories, and other sensor signal records—into a unified metadata schema, as illustrated in Figure 5.

For video preprocessing, the original fisheye recordings are undistorted into rectilinear frames to ensure compatibility with AV-LLMs. In scenarios with two wearer-mounted camera streams, the videos are temporally aligned to form a unified timeline. This alignment supports consistent segmentation of speech into sentences and facilitates accurate speech topic extraction.

## C.2 Annotation and Ground Truth Generation

Our annotation focuses primarily on objects and events.

**Static Object Annotation.** Static objects are automatically detected using EFM3D [61] based on a predefined list of object categories (e.g., couch, fireplace). We use Vision-LLM [41] to generate a informative description phrase for each detected object. Annotators then inspect the 3D coordinates and descriptions in a point-cloud viewer, correcting any errors in location, category, or description as needed.

**Sounding Event Annotation.** For each sound event, we annotate the event description or transcription, its start and end times, and the identity and 3D location of the sound source—if the source is tied to a physical object (e.g., running water with a faucet, a thud with a door). Human annotators adjust the event time span and label the source object and its position accordingly. Specifically for speech events, we first cluster raw word-level transcripts into complete sentences. Annotators then label speech events on a sentence-by-sentence basis. A prompted, rule-based agent [41] converts these validated sentences into concise speech topics that describe individual conversational moments. The prompt design used for this process is shown in Figure 6.

---

**Prompt: Word-Level Transcriptions to Speech Topic**

**[Task]**
You are an agent to annotate conversation data:

**[Rule]**
1. Create concise speech topics for each sentence that summarize what was said.
2. Use verb+ing format for all speech topics (e.g., `"Hello, how's it going."` $\rightarrow$ `"initiating conversation"`).
3. Ensure each speech topic is unique, using differentiating language for similar sentences.
4. Make topics concrete and specific enough that someone could identify the original sentence when hearing it.
5. Only reference what can be heard in audio (avoid visual elements like `"pointing"`).
6. Avoid abstract descriptions (e.g., use `"eating directly from bowl"` not `"announcing eating method"`).
7. Maintain the entire original CSV structure with all timestamps and durations.

**[Output]**
1. Add a `"speech_topic"` column right after the `"sentence"` column in the CSV.
2. Output Format: `rec_id,sentence,speech_topic,rec1_startTime,...etc.`

---

Figure 6: Prompt used to generate speech topics from word-level transcripts.

**Sound Event Annotation System and UI.** To streamline the annotation process and reduce errors, we developed a desktop annotation tool using PyQt5. This system integrates video playback, speech and non-speech event labeling, and timestamp editing in a single interface (see Figure 7). It supports dual-camera views with synchronized playback and saves annotations locally. The tool is self-contained, works offline, and requires no server backend.

**Human Annotation Guideline for Sound Events.** Annotators follow five key principles:

1) *Accuracy:* For speech events, correct the original word-level transcription to ensure that every spoken word and audible event is captured exactly as heard. Remove filler words and non-informative tokens, retaining only meaningful content.

2) *Completeness:* Label the full audible span of each event, setting start and end times as close as possible to the actual boundaries to avoid clipping or omission.

3) *Synchronization:* For speech events involving two participants, maintain the temporal alignment between the recordings from both devices throughout the annotation process.

4) *Label Uniformity:* For non-speech sound events, ensure that each description is unique, unambiguous, and consistent across the entire video.

5) *Language and Mechanics:* Use standard spelling, punctuation, and capitalization. Maintain consistent formatting across all annotations.

## C.3 QA Synthesis

We use template scripts to generate QA pairs for SAVVY-Bench. These scripts integrate the unified metadata (described in Section C.1) with the new annotations and ground truth data (from Section C.2) using well-defined question schemas, resulting in unambiguous and structured QA pairs.

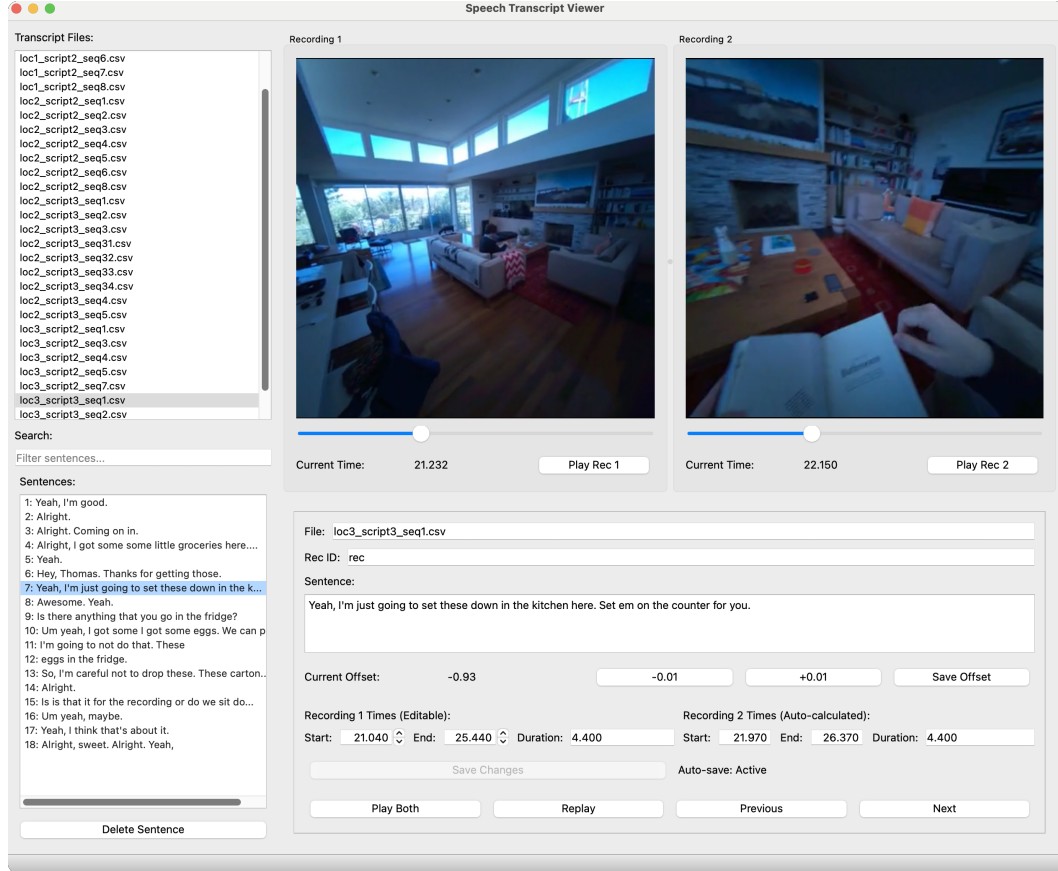

Figure 7: **Interface for sound event annotation.** The tool displays dual-camera videos with synchronized playback and saves annotations locally.

SAVVY-Bench includes six templates covering four task types: egocentric direction, egocentric distance, allocentric direction, and allocentric distance. For both egocentric and allocentric direction tasks, we design two levels of difficulty: a simple template with three options (left, right, back) and a hard template with four options (front-left, front-right, back-left, back-right).

We provide the complete set of templates for all six QA types, each specified for both speech and non-speech sound events as follows:

---

**Egocentric Direction - Simple**

1. Imagine you are the camera wearer, when the `{non-speech sound event}` sound comes up, relative to where you are facing, where is the sound source: left, right, or back? If the object is generally to your left and facing it requires turning less than 120 degrees left, choose 'left'. If the object is generally to your right and facing it requires turning less than 120 degrees right, choose 'right'. If the object is generally behind you and facing it requires turning 120 degrees or more, choose 'back'.

2. Imagine you are the camera wearer, when the speech topic `{speech topic}` comes up, relative to where you are facing, where is the other person : left, right, or back? If the object is generally to your left and facing it requires turning less than 120 degrees left, choose 'left'. If the object is generally to your right and facing it requires turning less than 120 degrees right, choose 'right'. If the object is generally behind you and facing it requires turning 120 degrees or more, choose 'back'.

---

## Egocentric Direction - Hard

1. Imagine you are the camera wearer, when the `{non-speech sound event}` sound comes up, relative to where you are facing, where is the sound source: front-left, front-right, back-left, or back-right? The directions refer to the quadrants of a Cartesian plane (if you are standing at the origin and facing along the positive y-axis). Consider the center point location of the object as the its location.

2. Imagine you are the camera wearer, when the speech topic `{speech topic}` comes up, relative to where you are facing, where is the other person: front-left, front-right, back-left, or back-right? The directions refer to the quadrants of a Cartesian plane (if you are standing at the origin and facing along the positive y-axis). Consider the center point location of the object as the its location.

## Egocentric Distance

1. Imagine you are the camera wearer, when the `{non-speech sound event}` sound comes up, relative to where you are standing, what is the distance between you and the sound source in meters? Consider the center point location of the object as the its location. Calculate the Euclidean distance between the two points in the horizontal plane. Answer in numeric format.

2. Imagine you are the camera wearer, when the speech topic: `{speech topic}` comes up, relative to where you are standing, what is the distance between you and the other person in meters? Consider the center point location of the object as the its location. Calculate the Euclidean distance between the two points in the horizontal plane. Answer in numeric format.

## Allocentric Direction - Simple

1. Imagine you are a robot standing by the `{reference object}` white recessed fireplace and facing `{facing object}`, when the `{non-speech sound event}` sound comes up, relative to where you are facing, where is the sounding object: left, right, or back? If the object is generally to your left and facing it requires turning less than 120 degrees left, choose 'left'. If the object is generally to your right and facing it requires turning less than 120 degrees right, choose 'right'. If the object is generally behind you and facing it requires turning 120 degrees or more, choose 'back'.

2. Imagine you are a robot standing by the `{reference object}` and facing the `{facing object}`, when the speech topic: `{speech topic}` comes up, relative to where you are facing, where is the speaker: left, right, or back? If the object is generally to your left and facing it requires turning less than 120 degrees left, choose 'left'. If the object is generally to your right and facing it requires turning less than 120 degrees right, choose 'right'. If the object is generally behind you and facing it requires turning 120 degrees or more, choose 'back'.

## Allocentric Direction - Hard

1. Imagine you are a robot standing by the `{reference object}` and facing the `{facing object}`, when the `{non-speech sound event}` sound comes up, relative to where you are facing, where is the sounding object: front-left, front-right, back-left, or back-right? The directions refer to the quadrants of a Cartesian plane (if you are standing at the origin and facing along the positive y-axis). Consider the center point location of the object as the its location.

2. Imagine you are a robot standing by the `{reference object}` and facing the `{facing object}`, when the speech topic: `{speech topic}` comes up, relative to where you are facing, where is the speaker: front-left, front-right, back-left, or back-right? The directions refer to the quadrants of a Cartesian plane (if you are standing at the origin and facing along the positive y-axis). Consider the center point location of the object as the its location.

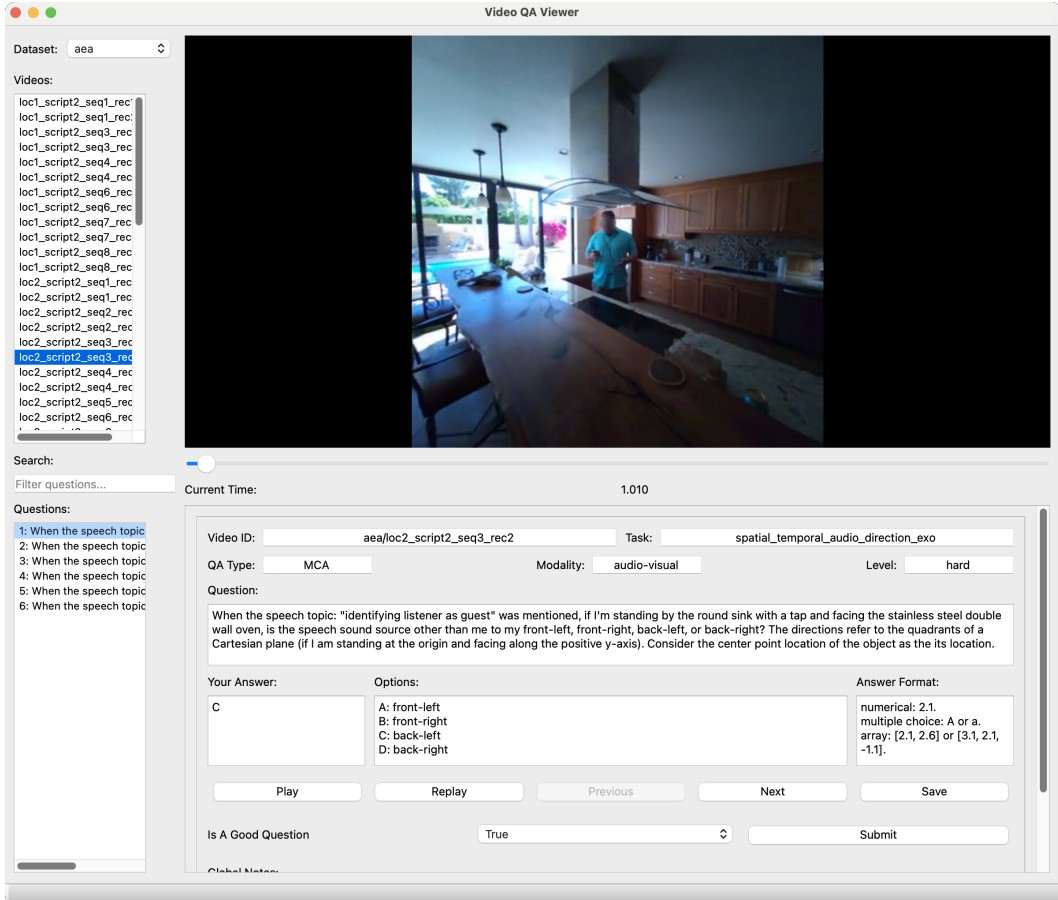

Figure 8: **Review interface for QA pair quality review.** The tool displays each video clip alongside its associated question and predicted answer, allowing reviewers to efficiently assess correctness, clarity, and formatting, and make a decision on whether the QA pair is a good QA that should be retained.

---

**Allocentric Distance**

1. When the `{non-speech sound event}` sound is happening, what is the distance between the `{reference object}` and the sounding object in meters? Consider the center point location of the object as the its location. Calculate the Euclidean distance between the two points in the horizontal plane. Answer in numeric format.

2. When the speech topic: `{speech topic}` is mentioned, what is the distance between the `{reference object}` and the speech sound source in meters? Consider the center point location of the object as the its location. Calculate the Euclidean distance between the two points in the horizontal plane. Answer in numeric format.

---

## C.4  Quality Review

We combine automated QA generation with manual review to ensure both scalability and quality. This hybrid pipeline enables efficient creation of large-scale QA pairs while preserving high annotation accuracy. The resulting dataset offers a reliable benchmark for evaluating 3D spatial reasoning in AV-LLMs. In this section, we detail the human quality review process that supports this workflow.

**Review Interface.** To secure the final data quality, we construct a review system with PyQt 5. The system presents each video clip together with its question and answer and offers a simple interface for reviewers to validate or revise the pair efficiently, as illustrated in Figure 8.

**Review Guideline.** Reviewers follow five principles:

1) *Correctness:* The stored answer must be fully supported by what is visible and audible in the clip.

2) *Clarity:* The question text must be clear and free of ambiguity.

3) *Relevance:* A question must refer only to content that is explicitly present in the clip or its metadata. It should not rely on commonsense inference or assumptions beyond what is observable.

4) *Consistency:* Answers must respect the predefined format, units, and option labels.

5) *Traceability:* Each reviewed QA pair is labeled as accepted or rejected based on whether it qualifies as a "good" question. All edits are logged to support future auditing and reproducibility.

# D    SAVVY-Bench Evaluation Details

## D.1    Open-Source AV-LLMs

All experiments are run in inference mode without model training. For open-source AV-LLMs at around 7B scale, we use a single A100 GPU (40GB). For 13B scale AV-LLM, we use a single 80GB VRAM A100 GPU. Evaluation follows the LMMs-Eval module [71]. We use greedy decoding with temperature set to 0, and both top-p and top-k set to 1. Following [71], we sample 32 video frames uniformly across the entire video duration. For audio, we average multiple channels to produce a compressed monaural input, with a sampling rate of 16kHz.

The input for the models is formatted as **[Video Frames], [Audio Content] and [Prompt]**

Prompt details:

---
**Relative Direction Questions - simple**

**[Question]**
Options: A: left B: right C: back.
Answer in single letter or numeric format.

---

**Relative Direction Questions - hard**

**[Question]**
Options: A: front-left B: front-right C: back-left D: back-right.
Answer in single letter or numeric format.

---

**Relative Distance Questions**

**[Question]**
Answer in single letter or numeric format.

---

## D.2    Proprietary Models

For Gemini-2.5-flash and Gemini-2.5-pro, we use Google Cloud Platform's API. We upload and feed the full video with audio to the model, following API guidelines.

Prompt details:

Given the Video: **[Video Frames]**,
Question: **[Question]**,
Options: **[Options]**

**[Prompt]**
Answer the question.

**[Format Instructions]**

1. Your output **must** be a single, valid JSON object conforming to the schema defined below.

2. **Do NOT** output any thinking steps or reasoning steps.

**[JSON Schema]**

```
{
   "prediction": "Your final answer (A, B, C, or A, B, C, D, or
      numeric value). If you can't decide, please output a JSON with
      the "prediction" key's value being null."
}
```

## D.3 Human Evaluation Guidelines

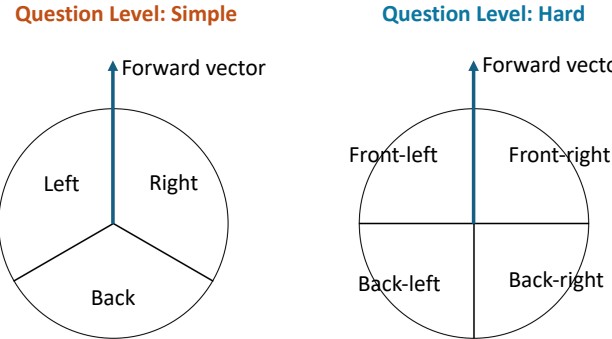

Figure 9: Direction quadrant guide for human evaluation. Egocentric directions are relative to the camera wearer's facing direction, while allocentric directions use a fixed world frame.

**Evaluation Setup.** We recruited six independent evaluators to participate in the human evaluation. The question set was shuffled and evenly divided among the evaluators. Each evaluator was allowed to pause, replay, or scrub through the video clip as many times as needed before submitting their answer. For direction-based tasks, evaluators followed the quadrant chart shown in Figure 9. For distance-based tasks, the correct response corresponds to the Euclidean distance between the two referenced points projected onto the horizontal plane.

**Evaluation Rules.** Evaluators followed four key rules:

1. *Perspective:* Identify whether the question requires egocentric or allocentric reasoning, and apply the appropriate frame of reference.

2. *Exactness:* Select the most accurate answer supported by visual and audio evidence, avoiding reliance on commonsense inference.

3. *Consistency:* Use the labels and answer formats provided (e.g., A, B, C or numerical values in the specified format).

4. *Independence:* Do not use any external tools such as object trackers or scene maps; rely solely on the provided video clip.

**Ethical Statement.** Participation was voluntary, involved no known physical or psychological risks, and did not collect any personal data beyond the evaluators' responses.

# E    Input Data Details

## E.1    Visual Input Settings

Original fisheye videos from the Aria-Everyday Activities (AEA) dataset [60] were undistorted to a standard rectilinear format for compatibility with common AV-LLM inputs. We also manually aligned the two camera-wearer videos for each conversation, creating a unified timeline to facilitate consistent speech sentence segmentation and speech topic generation. For all open-source AV-LLMs, we evaluated using 32 sampled video frames via uniform sampling.

## E.2    Microphone Configuration

We detail the microphone geometric configuration used in the AEA dataset. The data is collected using Meta's Aria Glasses, which are equipped with a 7-channel 48kHz microphone array distributed around the frame. Specifically, five microphones are positioned along the front frame, and two are mounted near the rear temple arms. This configuration enables rich spatial audio capture from both forward- and backward-facing directions.

The specific microphone locations (in meters, relative to the center of the glasses) are as follows:

- **Mic 0**: right-front-bottom corner $(0.05, -0.04, 0.00)$
- **Mic 1**: centered at the bridge of the nose $(-0.005, 0.00, 0.00)$
- **Mic 2**: left-front-bottom corner $(-0.05, -0.04, 0.00)$
- **Mic 3**: far-left-up along the front frame $(-0.07, 0.00, 0.00)$
- **Mic 4**: far-right-up along the front frame $(0.07, 0.00, 0.00)$
- **Mic 5**: rear left leg $(-0.07, 0.00, -0.10)$
- **Mic 6**: rear right leg $(0.07, 0.00, -0.10)$

A visualization of this microphone configuration is available on the Project Aria Hardware Specifications GitHub page.

## E.3    Camera Trajectory

SAVVY uses 6DoF camera trajectories at 1kHz. These trajectories approximate the continuous motion of the egocentric observer and are computed using the foundational visual-inertial odometry (VIO) and simultaneous localization and mapping (SLAM) systems onboard the Project Aria device. In our work, we use the calibrated closed-loop trajectories, represented by 3D position and orientation in quaternion form.

# F    SAVVY Details

## F.1    Snapshot Descriptor

As described in the main paper, the Snapshot Descriptor aims to: (1) identify the start and end times of the event; (2) determine whether the question requires an egocentric or allocentric view; (3) identify the *target sounding object*, *reference object*, and *facing object*, along with their text descriptions; and (4) track the egocentric direction and distance of each object at key frames.

To distinguish between views:

- **Egocentric view** refers to the camera wearer's perspective. In this case, the reference object is the camera, and no facing object is needed. Since the camera trajectory is known, only the target sounding object needs to be identified and tracked.

---

**Prompt: Open-Source AV-LLMs**

**[Task]**
Analyze the given video based on the question: "question". The total video length is duration seconds. Identify the **Sounding Object** (source of sound). Identify the **start_time** and **end_time** of the event mentioned in the question. Determine the mode:

- If I'm in the **camera wearer's view** (egocentric), set mode to `egocentric`.

- If I'm in a **different perspective** rather than the camera's view (allocentric), set mode to `allocentric`.

**[Output]**
Return a single JSON object with the following structure:

```
{
    "start_time": //start time of the event asked in the question
    "end_time": //end time
    "mode": egocentric/allocentric,
    "sounding_object": {
        "description": "A detailed description of the sounding object (
            source of sound). Include physical characteristics like type,
             color, material, and approximate size/shape.",
        "is_static": true/false // True if the object is generally non-
            moving, false if it typically moves location
    },
    "stand_by_object": {
        "object_name": "Name", //set to camera if requires_allocentric
            is false
        "description": "Description"
    },
    "facing_direction": {
        "object_name": "Name",
        "description": "Description"
    }
}
```

---

Figure 10: Prompt for Open-Source AV-LLMs on SAVVY-Bench.

- **Allocentric view** requires a perspective other than the camera's. A new coordinate frame is built using the reference object (as the origin) and the facing object (defining the positive y-axis from the reference). In this case, all three objects must be identified and accurately tracked.

Open-source AV-LLMs, typically at the 7B or 13B scale, often struggle to track all objects through prompt guidance. Therefore, we request these AV-LLMs to perform only the first three objectives: identifying the event time span, determining the view mode, and generating accurate object descriptions in correct object categories (target sounding / reference / facing object). For all models, we use greedy decoding with temperature set to 0, and both top-p and top-k set to 1.

Detailed prompts used for both open-source AV-LLMs and proprietary models to generate Snapshot Descriptor are provided in Figure 10 and 11 respectively.

## F.2 Text-Guided Snapshot Segmentation

We uniformly sample 128 frames from each video. For each object, we use its descriptive phrase, extracted from the Snapshot Descriptor, as input to ClipSeg [63] to generate a segmentation mask. Within the segmented region, we sample 10 keypoints and compute the average ClipSeg confidence. A detection is considered valid if the average score exceeds a threshold: 0.5 for dynamic sounding objects and 0.6 for reference and facing objects. We then use the selected keypoints and object descriptions to prompt the SAM model [64], obtaining refined segmentation masks.

**Prompt: Proprietary Models**

**[Task]**

Analyze the video at `uploaded_obj` based on the question: `question`.

Identify the **Sounding Object**, the **Reference Object**, and the **Facing Object** (stand by the **Reference Object** and face the **Facing Object**).

Identify the **start_time** and **end_time** of the event mentioned in the question.

Determine the mode:

- If I am in the **camera's view** (egocentric), set `mode` to `egocentric`.

- If I am in a **different perspective** rather than the camera's view (allocentric), set `mode` to `allocentric`.

Perform **audio-visual tracking** for these objects throughout the *entire duration* of the video.

**[Tracking Data]**

- For each object, provide its estimated position over time.

- Record positions at key moments across the *full video timeline* when the object is clearly visible in the frame.

- Estimate distance in meters from the camera to the object center.

- Estimate direction in degrees (−90 left to 90 right, 0 forward) from the camera.

**[Output]**

Your complete and sole output must be a single JSON object with the following structure:

```
{
    "event": "Brief description of the event from the question",
    "start_time": "minutes:seconds",
    "end_time": "minutes:seconds",
    "mode": egocentric/allocentric,
    "sounding_object": {
        "description": "A detailed description of the sounding object.
            Include physical characteristics like type, color, material,
            and approximate size/shape.",
        "is_static": true/false, // Set to true if the object is generally
            non-moving (like furniture, walls) and false if it typically
            moves location (like a person, animal, vehicle).

        "key_frames": { //*entire video* key visible frames
            "minutes:seconds": {"distance": "meters", "direction": "degrees
                "}
        }
    },
    "reference_object": { // Stand by Reference Object or camera
        "object_name": "Name",
        "description": "Description",
        "key_frames": { //*entire video* key visible frames
            "minutes:seconds": {"distance": "meters", "direction": "degrees
                "}
        }
    },
    "facing_object": { // Facing the facing_object, empty for camera
        "object_name": "Name",
        "description": "Description",
        "key_frames": { // *entire video* key visible frames
            "minutes:seconds": {"distance": "meters", "direction": "degrees
                "}
        }
    }
}
```

Figure 11: Prompt for Proprietary AV-LLMs (Gemini 2.5 models) on SAVVY-Bench.

To evaluate the robustness of SAVVY with the text-guided segmentation module (*Seg*), we conduct ablation studies on the ClipSeg confidence threshold (*Seg thr*) and the number of sampled frames (*N_frame*). We report sounding object localization accuracy (*loc_acc*) and QA accuracy on both egocentric and allocentric tasks from SAVVY-Bench. See the Experiments section of the main paper for detailed metric definitions.

For *Seg thr*, we test values 0.3, 0.5, 0.7, and 0.9, using the average ClipSeg score across keypoints, with all valid detections required to have at least one keypoint above 0.5. Results in Table 7 show stable performance across thresholds 0.3 to 0.7, with less than 3% variation. Lowering the threshold increases object recall, which improves sounding object localization accuracy (*loc_acc*), as SAVVY's egotrack-based outlier filtering and aggregation can effectively leverage the additional recalled samples. For QA tasks, a 0.5 threshold yields the highest overall accuracy, while 0.3 improves distance-related QA but reduces directional accuracy.

For *N_frame*, we evaluate 8, 16, 32, 64, and 128 frames (Table 8). Higher sampling rates lead to more valid detections from *Seg*, boosting sounding object *loc_acc* by 6.6% from 8 to 128 frames and improving egocentric QA accuracy. However, for allocentric QA, segmentation on static objects may introduce noise. As a result, lower frame counts like 32 or even 8 can perform comparably to 128 frames. These findings suggest a hybrid strategy: use *Seg* for sounding objects and rely more on other egotrack types such as the Snapshot Descriptor for static objects.

| Seg thr | Sound Loc loc_acc | Egocentric QA | | Allocentric QA | |
|---|---|---|---|---|---|
| | | direction | distance | direction | distance |
| 0.3 | 79.2 | 83.8 | 64.1 | 43.9 | 41.0 |
| 0.5 | 78.6 | 84.7 | 62.9 | 44.0 | 40.2 |
| 0.7 | 77.1 | 81.4 | 61.2 | 43.4 | 39.9 |
| 0.9 | 69.8 | 77.3 | 59.2 | 43.5 | 40.9 |

Table 7: Ablation results on the average snapshot segmentation confidence threshold (*Seg thr*). We report sounding object localization accuracy (*loc_acc*) and accuracy on egocentric and allocentric QA tasks. Lower thresholds generally yield higher sounding object recall, improving localization and distance-related QA accuracy with SAVVY, while moderate thresholds provide balanced performance.

| Seg N_frame | Sound Loc loc_acc | Egocentric QA | | Allocentric QA | |
|---|---|---|---|---|---|
| | | direction | distance | direction | distance |
| 128 | 78.6 | 84.7 | 62.9 | 44.0 | 40.2 |
| 64 | 76.7 | 82.7 | 61.6 | 43.0 | 40.2 |
| 32 | 74.8 | 81.9 | 61.1 | 43.7 | 41.4 |
| 16 | 73.8 | 81.9 | 59.8 | 43.2 | 40.5 |
| 8 | 72.0 | 80.1 | 59.4 | 44.7 | 39.9 |

Table 8: Ablation results on the number of sampled frames (*N_frame*) used in text-guided snapshot segmentation. Increasing the number of frames improves sounding object localization and egocentric QA accuracy. However, allocentric QA performance is less sensitive and can degrade at high frame counts due to noise in static object segmentation.

### F.3 Spatial Audio Cues

We process spatial audio signals at 0.25s per segment, with a sampling rate of 48 kHz. For each segment, we estimate the direction of arrival (DoA) by evaluating candidate angles over the full azimuthal range from $-180°$ to $180°$, sampled at $1°$ resolution. For each candidate angle, we apply the Generalized Cross-Correlation with Phase Transform (GCC-PHAT) method on each microphone pair to compute time-difference-of-arrival (TDOA) estimates. The angle $\hat{\phi}$ that maximizes the summed GCC-PHAT responses across all pairs is selected as the most likely direction of the source.

To assess the spatial diffuseness of the sound field for the sound source distance estimation, we compute the Coherent-to-Diffuse Ratio (CDR) from the multi-channel microphone signals. The input

to this process includes the raw microphone waveforms, the sampling frequency $fs$, microphone positions, and the estimated TDOAs for each pair. The analysis is constrained to the 500–2000 Hz frequency band for speech-related audio cues.

We estimate the power spectral densities (PSDs) and cross-spectral densities (CSDs) using Welch's method, with a segment length of 1536 samples (around 32ms) and 50% overlap. We clip negative values to zero and compute the mean CDR over the selected frequency band. The final CDR is averaged across all microphone pairs and serves as a global indicator of the ratio between coherent (direct-path) and diffuse (reverberant) components in the scene.

### F.4 Egocentric Track Aggregation

In the second stage of SAVVY, we aggregate three egocentric object tracks—produced by the Snapshot Descriptor, text-guided snapshot segmentation, and spatial cues—into a unified global map. Each per-frame trajectory is transformed into global coordinates, forming a global spatial map for downstream reasoning. The target object forms a time-varying global trajectory $\{\mathbf{p}_{\text{sound}}(t) \mid t \in \mathcal{T}_q\}$, while reference and facing objects are treated as static, with global positions $\mathbf{p}_{\text{ref}}$ and $\mathbf{p}_{\text{face}}$ computed by averaging their per-frame locations. These together define the **dynamic global map**:

$$\mathcal{M}_q = \{\mathbf{p}_{\text{sound}}(t) \mid t \in \mathcal{T}_q\} \cup \{\mathbf{p}_{\text{ref}}, \mathbf{p}_{\text{face}}\}\,.$$

We describe the aggregation strategies for static and dynamic objects below.

**Static objects.** Since the Snapshot Descriptor (SD) are better at localizing static objects (reference/facing) after track aggregation based on our ablation results (see main paper ablations), we prioritize the SD track. If the SD captures the object, we apply DBSCAN clustering (maximum distance of 1 m) on the SD track to determine a stable location. If the SD fails to detect the object, we fall back to the text-guided segmentation-based track (Seg), and apply DBSCAN with the same clustering threshold.

**Dynamic sounding object.** The Seg method is more accurate for tracking sounding objects (see main paper ablations), so we prioritize its trajectory when aggregating dynamic sound source tracks. We log Seg-tracked positions at each timestamp. For timestamps not covered by Seg, we query the SD track and filter outliers based on spatial consistency with the existing Seg trajectory. The resulting track is then extended by spatially fitting a smooth trajectory and removing outliers through the Seg-tracked points.

We then incorporate spatial audio cues to refine this trajectory. Specifically, we define a frustum-based search region for audio tracks around the target direction and distance, spanning a distance range of $\pm 1$ meter and an angular span of 45 degrees. We sample candidate points at the centers of 10 angular bins and 5 distance bins within this region. If the audio indicates that the object is located behind the camera (i.e., absolute angle $\theta > 90°$), or provides positional information for timestamps not covered by Seg or SD, we refine the track by comparing with audio-based predictions. Inconsistent points are filtered based on spatial agreement with nearby audio-informed estimates, and the trajectory is extended accordingly to produce the final track.

The aggregation process can be summarized as Algorithm 1.

**Discussion: What roles does the global mapping play in SAVVY?**

Camera trajectory serves as the bridge between Stage 1 egocentric tracks and the Stage 2 dynamic global map. It can be obtained using real-time SLAM technologies [62, 60] with devices such as AR glasses or robotic sensors. Given camera pose (location and orientation), egocentric direction $\theta$ and distance $r$ can be transformed into global 3D coordinates. This transformation allows tracks from multiple modalities—Snapshot Descriptor (*SD*), text-guided snapshot segmentation (*Seg*), and spatial audio cues (*Audio*)—to be aligned in a shared 3D coordinate system (global mapping). Different modalities may capture object trajectories at different timestamps; by mapping them to a global frame, these partial observations can complement each other. Through outlier filtering and temporal smoothing, we obtain reliable tracks for dynamic objects and stable positions for static ones.

Table 9 compares performance with and without global mapping in terms of sounding object localization accuracy (*loc_acc*) and egocentric QA accuracy (*direction* and *distance*) on SAVVY-Bench. In the *w/o Global Mapping* setting, we directly take egocentric tracks from SD, Seg, and Audio based on the Snapshot Descriptor's grounded time span, then vote on direction and take the median angle and

---

**Algorithm 1** Track Aggregation Algorithm for Global Map Construction

---

1: **Input:** $\mathcal{S}, \mathcal{D}, \mathcal{A}$ (dense segmentation, SD, audio tracks); $o$ (object type); $\mathbf{L}(t)$ (camera trajectory); $\mathcal{T}_q$ (query time range)

2: **Define:** $\texttt{MapToGlobal}(\boldsymbol{\tau}, \mathbf{L}(t)) := \mathbf{L}(t) + \begin{bmatrix} r \cdot \cos(\theta) \\ r \cdot \sin(\theta) \end{bmatrix}$, where $\boldsymbol{\tau} = (t, \theta, r)$

3: Initialize map $\mathcal{M}_q \leftarrow \varnothing$

4: **if** $o$ is static **then**

5:     **for** each $\boldsymbol{\tau} \in \mathcal{D}, \mathcal{S}$ **do**

6:         $\mathbf{p}(t) \leftarrow \texttt{MapToGlobal}(\boldsymbol{\tau}, \mathbf{L}(t))$

7:         **break**

8:     **end for**

9:     $\bar{\mathbf{p}} \leftarrow$ centroid of clustered $\mathbf{p}(t)$

10:     $\mathcal{M}_q \leftarrow \mathcal{M}_q \cup \{\bar{\mathbf{p}}\}$

11: **else**

12:     Initialize trajectory $\mathbf{p}(t) \leftarrow \varnothing$

13:     **for** each $t \in \mathcal{T}_q$ **do**

14:         **for** each $\boldsymbol{\tau}$ in $\{\mathcal{S}, \mathcal{D}, \mathcal{A}\}$ if $t \in \boldsymbol{\tau}$ **do**

15:             Filter outliers near $\mathbf{p}(t')$

16:             $\mathbf{p}(t) \leftarrow \texttt{MapToGlobal}(\boldsymbol{\tau}, \mathbf{L}(t))$

17:         **end for**

18:     **end for**

19:     Interpolate and smooth $\mathbf{p}(t)$ over $\mathcal{T}_q$

20:     $\mathcal{M}_q \leftarrow \mathcal{M}_q \cup \{\mathbf{p}(t)\}$

21: **end if**

22: **return** $\mathcal{M}_q$

---

distance at the queried time. Global mapping improves single-modality performance, especially for dense tracks like Seg and Audio, which see localization accuracy (*loc_acc*) gains of about 10%. SD, being sparse, is less sensitive to global mapping and may perform better without it. For combined modalities, global mapping not only supports self-correction within each modality but also enables cross-modality completion, yielding even greater improvements—up to 11.5% on egocentric distance accuracy and *loc_acc*. Full SAVVY with all three tracks shows the strongest gains: +11.9% in *loc_acc*, +14.3% in egocentric distance accuracy, and +4.1% in direction estimation.

| Track Type | | | w/ Global Mapping (SAVVY) | | | w/o Global Mapping | | |
|:---:|:---:|:---:|:---:|:---:|:---:|:---:|:---:|:---:|
| SD | Audio | Seg | loc_acc | direction | distance | loc_acc | direction | distance |
| ✓ | | | 55.7 | 68.3 | 47.9 | 56.3 | 71.1 | 52.6 |
| | ✓ | | 59.0 | 73.9 | 48.1 | 49.7 | 75.6 | 40.1 |
| | | ✓ | 72.5 | 81.2 | 52.0 | 62.3 | 75.8 | 43.7 |
| ✓ | ✓ | | 66.8 | 74.5 | 54.6 | 55.3 | 73.0 | 43.3 |
| ✓ | ✓ | ✓ | 78.6 | 84.7 | 62.9 | 66.7 | 80.6 | 48.6 |

Table 9: Ablation study on the impact of global mapping. We evaluate combinations of egocentric track modalities—Snapshot Descriptor (*SD*), Spatial Audio (*Audio*), and Segmentation (*Seg*)—with and without global coordinate transformation. Metrics include sounding object localization accuracy (*loc_acc*) and egocentric QA accuracy on SAVVY-Bench (*direction* and *distance*). Global mapping consistently enhances performance, particularly when aggregating dense tracks (*Seg* and *Audio*) and integrating multiple modalities.

## G  Efficiency Analysis

We report average latency and peak GPU memory for each stage over 200 test samples on a single NVIDIA A100. **1) SLAM**: real-time on Aria glasses. **2) SD**: one AV-LLM forward pass per QA (latency comparable to standard AV-LLM inference). **3) *Audio & Global Map*:** fewer than $0.1$ s per sample on CPU. **4) *Seg*:** $\approx 0.52$ s/frame at $512 \times 384$ per object; peak GPU memory $9.4$ GB. Depth estimation costs $\approx 0.44$ s/frame (up to $6$ GB). While *Seg* is the primary bottleneck of the efficiency, Table 8 shows that using 32 frames per video for *Seg* maintains strong accuracy. Besides, removing *Seg* (Table 6 in main paper) substantially reduces runtime while still outperforming LLM-

only baselines, particularly on allocentric questions. Overall, modular design of SAVVY enables accuracy–efficiency trade-offs to meet real-time deployment constraints.

## H Blind Testing

We conduct blind testing to evaluate the contribution of the visual modality in audio-visual spatial reasoning on SAVVY-Bench, using AV-LLM baseline models. Specifically, we compare performance between two settings: *Audio Only* (removing visual frames, using only audio and the text query as input) and *Audio + Visual* (using both modalities). We evaluate on egocentric QA tasks to assess how models infer the direction and distance of sound sources relative to the camera.

We test the top five open-source 7B models and the strongest proprietary model, Gemini-2.5-pro. As shown in Table 10, Gemini demonstrates strong grounding capabilities (67.4% t-mIoU, as reported in the main paper), and its performance shows a clear dependence on visual input. Under *Audio Only*, Gemini's direction accuracy drops sharply by 32.4%, while distance accuracy decreases by only 2.8%. This aligns with observations from our reasoning process visualizations: Gemini relies heavily on visual input for spatial direction reasoning, whereas distance estimation is less affected—likely due to the role of commonsense priors from audio and language.

Other AV-LLMs exhibit similar trends: direction accuracy degrades more under *Audio Only*, while distance accuracy remains relatively stable or even improves for some AV-LLMs such as MiniCPM-o. However, the performance gap is smaller than with Gemini, likely because these models fail to reliably ground events in time—achieving less than 5% t-mIoU—regardless of the input modality. As a result, even with visual input, their spatial reasoning remains limited.

| Method | Audio Only | | Audio + Visual | |
| --- | --- | --- | --- | --- |
| | Direction | Distance | Direction | Distance |
| VideoLLaMA2-7B [21] | 39.1 | 40.7 | 46.4 | 42.7 |
| MiniCPM-o 2.6 [25] | 41.9 | 50.7 | 45.8 | 42.3 |
| EgoGPT [23] | 39.3 | 37.0 | 40.2 | 57.6 |
| Gemini-2.5-pro | 42.8 | 56.8 | 75.2 | 59.6 |

Table 10: Blind testing on SAVVY-Bench: comparison between *Audio Only* and *Audio + Visual* input settings. Reported metrics are egocentric QA accuracy for direction and absolute distance. Gemini-2.5-pro shows the largest gap, indicating strong reliance on visual input for accurate direction estimation.

## I Limitations

One limitation of SAVVY is that it currently relies on a strong foundational AV-LLM—specifically Gemini—and inherits its capabilities in temporal grounding and object referral. The pipeline may underperform if the base model lacks these abilities in the initial stage. Additionally, the spatial audio tracking module uses rule-based signal processing: while effective for direction estimation, distance estimation remains challenging, particularly given the wide variance of near- and far-field cases in the current dataset. Future work could improve audio-visual track aggregation by enhancing this module through large-scale training on realistic spatial audio data.

## J Broader Impacts

This work contributes to the development of AV-LLMs capable of fine-grained spatial reasoning in dynamic 3D environments. By introducing a benchmark and training-free pipeline that enables structured spatial understanding across audio and visual modalities, our work opens new avenues for intelligent multi-modal systems in domains such as assistive robotics, AR/VR, human-computer interaction, and audio-visual navigation [72]. These capabilities have the potential to significantly enhance accessibility tools (e.g., guiding visually impaired users through complex spaces), improve AR/VR user experiences, and support more context-aware AI agents in embodied environments.

However, alongside these benefits, the increasing power of AV-LLMs introduces potential risks. Models capable of interpreting spatial relationships from audio-visual input could be misused in surveillance applications, unauthorized tracking, or context inference without user consent. Moreover, as our method builds on these foundation models, it inherits their limitations and biases, which can propagate through the pipeline and affect real-world deployments. There is also the risk that such models may make confident but incorrect spatial inferences in safety-critical settings. To mitigate these concerns, we recommend that future systems incorporating AV-LLMs for spatial reasoning include safeguards such as: (1) explicit transparency about model uncertainty and failure modes; (2) data collection and evaluation guidelines that prioritize privacy and ethical use of human-centered audio-visual data; and (3) usage restrictions for sensitive applications, especially those involving biometric data or real-time environmental monitoring. Furthermore, research into interpretability and robustness of spatial reasoning components will be critical for safe deployment.

## K    Additional Qualitative results: Reasoning Error Analysis

In this section, we show additional reasoning examples of Gemini-2.5-pro and conclude four major types of errors in the visualization:

1) *Referral Error:* This error occurs when the model fails to correctly identify, locate, or interpret the properties of specific objects, persons, or abstract reference points mentioned in the question. It is particularly common when the referenced object descriptions are complex, rely on relative positioning (e.g., "the armchair further from the wall painting"), or refer to abstract sound events (e.g., "a thud sound") that are not tied to a clearly visible object and must be inferred from broader video context. The model may select an incorrect referent or misinterpret its attributes, leading to a flawed premise for subsequent spatial reasoning. An example is shown in Figure 12, where the model incorrectly identifies the queried armchair (the facing object) as the one at the arched opening.

2) *Temporal Localization Error:* This error occurs when the model fails to accurately identify the correct time span of the queried sound event in the question. As a result, the model analyzes the spatial context at an incorrect point in time, leading to flawed reasoning about object locations or spatial relationships. Figure 13 shows an example where the model confuses the speech event "suggesting trying the coffee" with another semantically similar topic, "complimenting the coffee taste," leading to an error in egocentric direction prediction.

3) *Spatial Relationship Error:* This error occurs when the model misinterprets or misapplies fundamental spatial relationships (e.g., left/right, front/back, in front of/behind, next to, between) between correctly identified entities, even within a correct frame of reference. In Figure 14, the model successfully identifies the correct event time span, detects all relevant objects as well as their locations. However, it fails to interpret the relative direction correctly, placing the object on the right side of the robot's view instead of the left, resulting in an incorrect prediction of "front-right" rather than the correct "front-left."

4) *Spatial Measurement Error:* This error arises in tasks that require quantitative responses—such as estimating distances or making precise angular judgments (e.g., in Snapshot Descriptor-based tasks). Even when the model correctly identifies the relevant objects and understands their qualitative spatial relationships, it may still make significant errors in geometric reasoning (e.g., applying Pythagorean theorem incorrectly, flawed calculation logic), scale estimation, or numerical calculations. Figure 15 presents an example where the model correctly identifies both the sound source and the queried reference object, and even retrieves a relevant navigation path between them. However, it fails to calculate the distance accurately. This case also reveals a typical reasoning pattern in AV-LLMs for distance estimation: the model anchors the sound source and reference object to static landmarks in the scene, recalls the relevant navigation routes observed in the video, and then estimates the distance according to the routes.

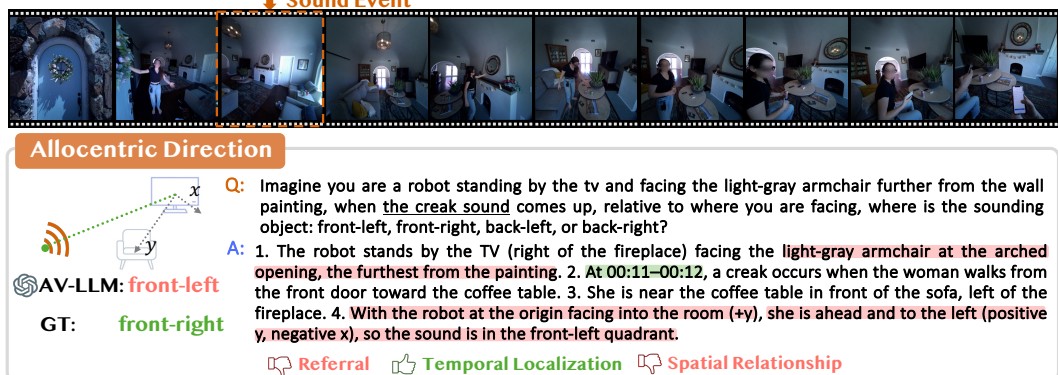

Figure 12: **Example of Referral Error.** The model incorrectly selects the armchair (facing object) at the arched opening, failing to identify the correct object description queried in the question. This misinterpretation further causes error for the spatial relationship.

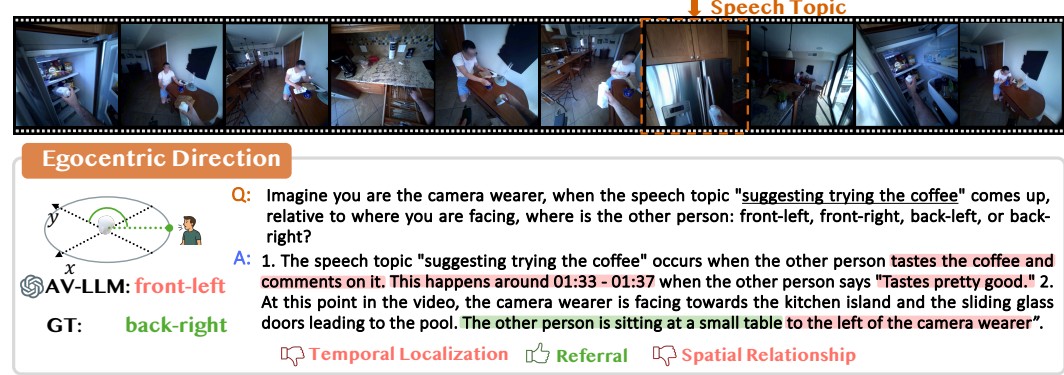

Figure 13: **Example of Temporal Localization Error.** The model incorrectly grounds the speech event "suggesting trying the coffee," confusing it with a similar topic. This mismatch causes incorrect egocentric direction prediction.

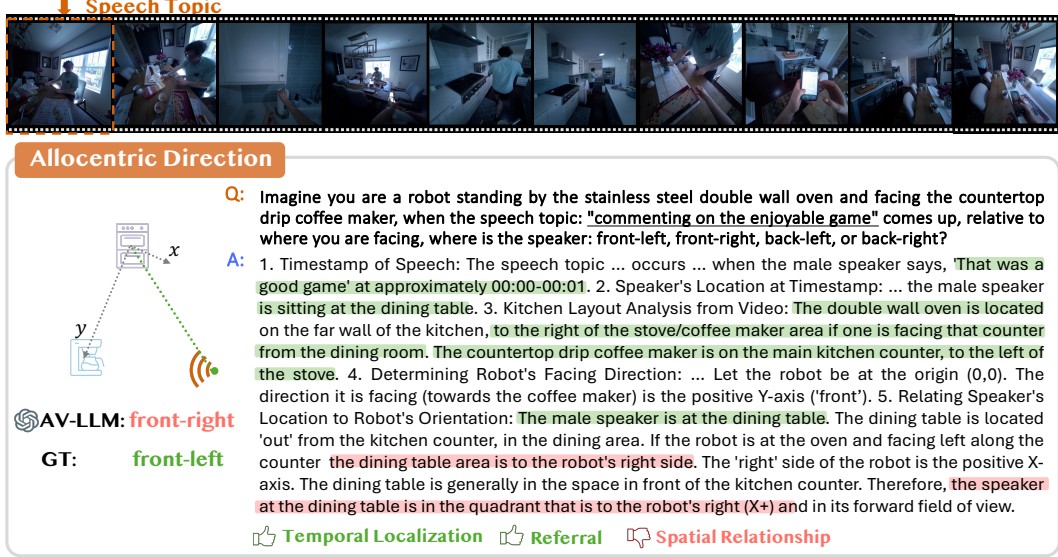

Figure 14: **Example of Spatial Relationship Error.** The model correctly identifies the queried time span, relevant objects, and their locations, but fails to determine the correct relative direction of the sounding object with respect to the robot's view. It incorrectly predicts the object as being on the right side instead of the left, leading to a misclassification of "front-right" instead of the correct "front-left."

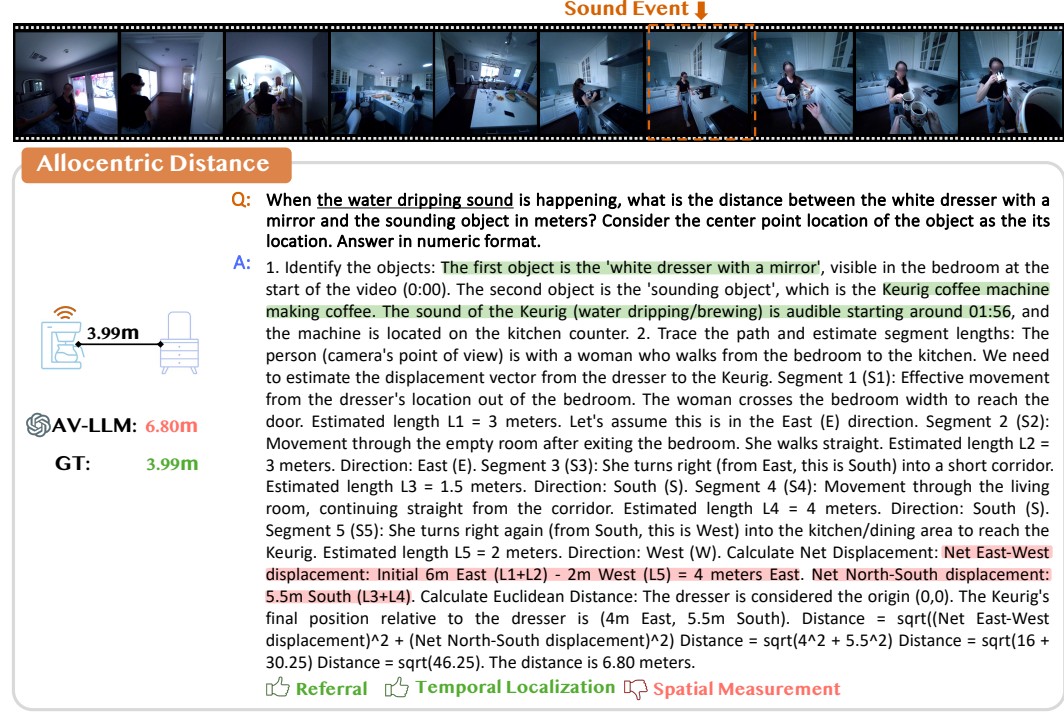

Figure 15: **Example of Spatial Measurement Error.** The model correctly identifies the sound source and reference objects, but fails to compute the distance accurately along with the navigation route from the reference object to the sound source.

