# OpenReview forum: "SAVVY: Spatial Awareness via Audio-Visual LLMs through Seeing and Hearing"
_NeurIPS.cc/2025/Conference — NeurIPS 2025 oral_

### Official Review · Reviewer_vh2k · 2025-06-08

**Clarity:** 3
**Significance:** 3
**Originality:** 3
**Rating:** 5
**Confidence:** 2

**Summary:**

This paper introduced a benchmark called SAVVY-bench that evaluate the dynamic audio-visual reasoning capabilities of models. They also introduced a training-free pipeline that augments existing AV-LLMs to achieve state-of-the-art performance on SAVVY-bench.

**Questions:**

1. what is the bottleneck for models on this benchmark at this point?

**Ethical Concerns:**

["NO or VERY MINOR ethics concerns only"]

**Quality:**

3

**Strengths And Weaknesses:**

Strengths:
1. audio-visual spatial reasoning through time has been a missing piece in LLM evaluation, and this paper does exactly that. The task construction is based on real world dataset rather than simulated environment, which make it more reliable.
2. The construction of the data is rigorously documented in main paper and supp. It's an impressive amount of work.
3. The proposed pipeline improve baseline Gemini 2.5 pro significantly without any training

Weakness:
1. Not really a weakness, but the approach SAVVY basically shows that compared to using LLM E2E, we get better results if we applied traditional methods in segmentation, mapping and localization to extract information (and using LLM also just as a feature extractor). It's hard to pin point the novelty in the approach, as it's a combination of existing methods.

---

> ### Author Rebuttal · Authors · 2025-07-31
>
> We thank the reviewer for detailed and helpful feedback and we appreciate that the reviewer acknowledges the uniqueness of our proposed task, and the real-world aspect of the constructed benchmark, and the effectiveness of the proposed reasoning pipeline. We address the questions and concerns below.
>
> **1. Novelty Clarifications.**
>
> We claim our novelty in the following two aspects on benchmark and reasoning framework:
> + **First-of-its-kind Benchmark:** SAVVY introduces a novel benchmark, pipeline, and reasoning structure grounded in multi-modal alignment in the 3D space. SAVVY-Bench is the first benchmark specifically designed for 3D spatial reasoning with synchronized spatial audio. Unlike existing AV benchmarks that focus on 2D video understanding or simple audio-visual correspondence, we introduce temporally-grounded, spatially-aware questions that require true 3D understanding.
>
> + **Novel Reasoning Framework**: While SAVVY combines existing tools, its core contribution lies in how these components are orchestrated to enhance spatial reasoning in AV-LLMs.
>
> SAVVY motivates and formalizes the integration of structured egocentric tracks (via the Snapshot Descriptor), spatial audio cues, and visual cues  into a shared global 3D map. This design allows us to study each modality’s contribution for the audio-visual spatial reasoning tasks.
>
> **Importantly, SAVVY not only improves performance over E2E LLM baselines, but also opens up new directions: building multi-step reasoning chains, distilling knowledge back into AV-LLMs, and eventually enabling end-to-end systems that reason more effectively, without relying on heavy modular components. Thus, the novelty lies in framing the right questions, designing a principled pipeline, and enabling new research paths in audio-visual spatial reasoning.**
>
> **2. Bottlenecks for Models on SAVVY-Bench.**
>
> We identify the following bottlenecks in current AV-LLMs on our benchmark:
> + **Limited Spatial Audio Understanding:** All models take mono audio as input and overlook the spatial audio cues for spatial audio reasoning tasks.
> + **Over-Reliance on Visual Cues:** Models often depend heavily on visual input, which becomes unreliable in out-of-view or occluded scenarios. As illustrated in Figure 4 (top) of Section 4.2, when a sound source disappears from view, models extrapolate from its last seen location, leading to incorrect spatial reasoning.
>
> **For open-source 7B models specifically, Table 3 shows two major bottlenecks:**
> + **Temporal Grounding:** Models achieve under 5% t-mIoU, reflecting poor event-time alignment. Synchronizing complex audio events like speech remains a challenge at this scale.
>
> + **Object Referral Accuracy:** For spatial reasoning, models must identify and describe relevant objects to establish spatial frames. However, they reach only ~33% accuracy, revealing limitations in visual-language grounding under spatial context.
>
> Looking ahead, we envision next-generation wearable devices with enhanced audio-visual capabilities that will make LLM-based spatial reasoning ubiquitous in daily life. By establishing SAVVY-Bench now, we’re laying the groundwork for this future – creating a bridge between high-quality multi-modal data collection and practical AI applications. Our goal is to catalyze a complete ecosystem where advances in data, models, and applications reinforce each other, ultimately transforming how humans interact with and understand their spatial environment.

---

### Official Review · Reviewer_tKUJ · 2025-07-02

**Clarity:** 4
**Significance:** 4
**Originality:** 4
**Rating:** 5
**Confidence:** 3

**Summary:**

The authors design a benchmark, SAVVY-Bench, to evaluate the spatial reasoning capabilities of audio-visual large language models (AV-LLMs). This is the first work to incorporate spatial audio as a condition in spatial perception tasks for LLMs. The benchmark evaluates existing AV-LLMs from both egocentric and allocentric perspectives, across both direction and distance dimensions. The authors also propose SAVVY, a training-free multimodal spatial reasoning system built on top of a state-of-the-art AV-LLM. SAVVY integrates three distinct spatial direction and distance estimation methods: Snapshot Descriptor, Text-Guided Snapshot Segmentation, and Spatial Audio Cues. The outputs from these three methods are denoised (outlier removal) and aggregated to construct egocentric spatial tracks, which are then used to build a Dynamic Global Map.

In experiments, the authors demonstrate that SAVVY outperforms all existing state-of-the-art AV-LLMs on the SAVVY-Bench benchmark. However, all AV-LLM systems, including SAVVY, still lag behind human performance in spatial perception tasks, indicating that there remains significant room for improvement in audio-visual spatial reasoning for LLMs. Ablation studies further validate that the three proposed methods for egocentric spatial track construction each contribute uniquely to different aspects of spatial perception. Moreover, the experiments show that combining all three methods yields the best overall performance in audio-visual spatial reasoning.

**Questions:**

1. In Table 6, the spatial reasoning capabilities of the combinations SD+Seg and Audio+Seg are not compared. It would be helpful if the authors could include results for these two cases. There is a possibility that one of the methods used for egocentric tracks construction might have a significant negative impact, causing the combination of all three methods to perform worse than the combination of just two methods.

2. In Table 5, the spatial direction and distance estimation performance of Audio is worse than SD. However, Table 6 shows that Audio outperforms SD in terms of sound localization accuracy and egocentric direction and distance estimation. Are these two results in conflict? Could the authors provide an analysis of this discrepancy?

3. Is there a computational overhead comparison between SAVVY and other methods? The pipeline combines SLAM, AV-LLMs, image segmentation, and audio processing. Latency/throughput metrics are absent, making real-time deployment feasibility unclear.

**Ethical Concerns:**

["NO or VERY MINOR ethics concerns only"]

**Limitations:**

1. The question types in SAVVY-Bench are relatively monotonous, with only 4 types of questions included.

2. The computational overhead of SAVVY could be significant because of the combination of so many foundation models.

**Quality:**

4

**Strengths And Weaknesses:**

a) Strengths
1. The authors are the first to introduce spatial audio into spatial reasoning for LLMs. From the perspective of multimodal spatial perception, integrating both audio and visual spatial cues aligns more closely with the way humans perceive space in real life. This task is both meaningful and novel.
2. The SAVVY-Bench constructed by the authors has undergone meticulous data preprocessing, annotation, QA synthesis, and quality review processes. The methodology used to build the benchmark is rigorous.
3. The proposed SAVVY system estimates spatial direction and distance from the vision perspective using SD and Seg methods, and from the audio perspective using existing spatial audio localization and distance estimation models. The system effectively removes outliers and integrates information from both vision and audio, making excellent use of both modalities and achieving performance that surpasses the state-of-the-art AV-LLMs.
4. In terms of both writing and experimentation, the authors provide comprehensive quantitative and qualitative analysis.

b) Weaknesses
1. The question types in SAVVY-Bench are relatively monotonous, limited to only egocentric/allocentric and direction/distance, resulting in the evaluation of a rather narrow spatial reasoning task.
2. The system is described as a training-free approach, which is a strength. However, this may come with computational trade-offs. It would be useful to discuss the computational cost of the SAVVY system in comparison to other state-of-the-art models, especially since it integrates multiple methods (e.g., Snapshot Descriptors, Segmentation, and Audio Cues). This could be particularly important for practical deployment in real-time systems.

---

> ### Author Rebuttal · Authors · 2025-07-31
>
> We are grateful to the reviewer for detailed and supportive feedback, and we are glad to see that the reviewer acknowledges the novelty of our work, the quality of the proposed benchmark, the effectiveness of SAVVY, and solid experiments along with detailed analysis. We address the questions and concerns below.
>
> **1. Limited Question Types.**
>
> The evaluated tasks aim to represent realistic interplay between audio-visual-language understanding. While more abstract reasoning (e.g., voice gender classification or crowd counting) is valuable, **our focus is on fundamental skills—such as direction, distance, temporal grounding and grounded object referral—that are critical to real-world applications and pave the way for broader reasoning capabilities**. During our work on this paper, we find that even such fundamental tasks, without further complexities, require novel approaches in multi-modal learning. We preferred to focus on these fundamental skills, proposing the first benchmark (SAVVY-Bench) as well as the approach (SAVVY).
>
> Noteworthy, we do introduce some amount of complexity into the benchmark. **Beyond direction and localization QAs, our benchmark also introduces Snapshot Descriptor Evaluation (SD-Eval) in Table 3 (Section 4.2)**, covering: **(i) Temporal Grounding** – measurement of how accurately a model would localize queried sound events in time. **(ii) Object Referral** – tests whether the model correctly describes objects given the question and the video. We also propose a new localization accuracy metric comparing predicted and ground truth positions to test SAVVY and its variants.
>
> Indeed, most 7B open-source models achieve <5% temporal mIoU, under 35% referral accuracy, and perform only slightly above chance on direction and distance QA—highlighting that current open-source AV-LLMs still struggle with temporal alignment, semantic grounding, and basic audio spatial reasoning. These limitations make it difficult for these models to be evaluated meaningfully on more complex tasks beyond what SAVVY-Bench currently covers.
>
> **2. SAVVY Complexity and Runtime Efficiency.**
>
> We acknowledge SAVVY’s multi-stage pipeline, but it is training-free and modular by design—allowing components to be adapted or replaced as needed. Each stage serves a targeted role in spatial reasoning (e.g., segmentation for dynamic object tracking, audio cues for out-of-view events) and also lays the groundwork for future research on reasoning knowledge distillation and chain-of-thought in AV-LLMs.
>
> In regards to runtime efficiency, we include detailed latency and peak memory usage metrics below (will also be incorporated in revised version of the manuscript):
>
> The time and memory costs (where applicable) for each step of the SAVVY pipeline are:
> + SLAM: real-time slam on device (aria glasses).
> + Stage 1 Step 1 (Snapshot Descriptor): Runs the AV-LLM once per QA to determine temporal spans and object roles—this matches standard AV-LLM inference latency on our benchmark (Table 2 and 3).
> + Stage1 Step 2 (Text-Guided Snapshot Segmentation with Depth Estimation):
> Segmentation (CLIPSeg/SAM2) is the most time-consuming (~0.52s for one frame at 512×384 per object), but we show in Supp. Table 2 that using just 32 frames uniformly sampled in one video still yields strong performance.
> Depth estimation takes ~0.44s per frame and ~6GB GPU memory; segmentation up to ~9.4GB.
> Speed and memory were tested on one NVIDIA A100 GPU, average on 200 testing samples.
> + Stage1 Step 3 (Spatial Audio Cues): Lightweight—0.06s per second of audio, CPU-only.
> + Stage 2 (Global Map Construction): Fast and efficient—0.08s per video, CPU-only.
>
> Notably, the most time-consuming stage is segmentation, while other steps—such as audio processing and global mapping—are lightweight and run efficiently on CPU. Without segmentation, the total latency is comparable to a single AV-LLM inference (Table 6), and the pipeline remains feasible for near real-time use in controlled settings.
>
> We view SAVVY as a modular and extensible framework rather than a fixed system—paving the way for real-time, efficient adaptations in future work.
>
> **3. On Missing SD+Seg and Audio+Seg Results in Table 6.**
>
> Thank you for the suggestion to add results of SD+Seg and Audio+Seg —we have added the missing combinations in Table 6:
>
>
> | Track Type | loc_acc | Ego Dir | Ego Dist | Alloc Dir | Alloc Dist |
> |------------|---------------|---------|----------|-----------|------------|
> |SD+Seg|76.0|83.2|54.4|44.2|36.9|
> |Audio+Seg|73.6|84.2|57.7|34.4|24.0|
>
>
> Together with other results in Table 6, these findings highlight the complementary strengths of each modality:
> + **Text-guided segmentation significantly contributes to localizing dynamic sounding objects**, improving direction grounding, especially for egocentric questions.
> + **Spatial audio cues enhance distance estimation largely and also improve both localization and direction QA accuracy**.
> + **Snapshot Descriptor (SD) performs best on allocentric reasoning by more accurately localizing reference and facing objects** to construct meaningful spatial relationships.
> + Importantly, **combining all three modalities does not introduce negative interference.**
> These results underscore the value of integrated reasoning across modalities for robust spatial understanding.
>
>
> **4. On the Discrepancy Between Table 5 and Table 6 (Audio vs SD).**
>
> We are glad to explain the interpretation of Table 5 and Table 6. **The results in Table 5 and Table 6 are not contradictory—they reflect evaluations under different settings**:
> + Table 5 evaluates direction and distance estimation accuracy for egocentric tracks **without global trajectory mapping**.
>
> + Table 6 shows results **after global mapping and trajectory integration**, where audio cues that are denser and more continuous can be accumulated and refined over time, leading to stronger performance than SD after integration.
>
> Both SD and Audio localization accuracy improve with global mapping and trajectory integration. **This highlights the value of global mapping, which aligns egocentric tracks into a shared 3D coordinate frame, which enables the fusion of observations into reliable dynamic object tracks and stable spatial references** (see detailed discussion in lines 277–297 of the supplementary PDF).

---

### Official Review · Reviewer_cjrp · 2025-07-02

**Clarity:** 3
**Significance:** 3
**Originality:** 3
**Rating:** 5
**Confidence:** 3

**Summary:**

This paper introduce the the first benchmark for 3D spatial reasoning in dynamic scenes with synchronized spatial audio: SAVVY-Bench, and and a fully training-free two-stage pipeline that leverages off-the-shelf components
*(pretrained audio-visual LLMs for timestamped snapshot extraction, segmentation and monocular depth networks for object localization, SRP-PHAT/CDR for audio direction-of-arrival and distance estimation, SLAM-based camera poses for global mapping, DBSCAN clustering for static object centroids, and Kalman filtering for dynamic trajectory smoothing)*
to accurately answer “which way?” and “how far?” questions in moving egocentric video with spatial audio, all without any additional model fine-tuning.

**Questions:**

- Can the author report end-to-end latency (per QA) and peak memory/GPU utilization on representative hardware?
- Has the author tested performance on non-Aria or outdoor recordings? It would be great if the author could report the metrics.
- Can the author quantify the accuracy of your zero-shot LLM-based event timestamps? I feel like this misalignment here will affect downstream spatial accuracy.

**Ethical Concerns:**

["NO or VERY MINOR ethics concerns only"]

**Final Justification:**

My final recommendation is to Accept.

My initial concerns about the pipeline's efficiency and the benchmark's limited scope were resolved by the authors' rebuttal.

- Key Issues Resolved: The authors provided the latency and memory metrics, confirming the method's practical feasibility.
- Remaining Limitations: The use of indoor-only data is an acknowledged limitation but is acceptable for a paper establishing a novel benchmark.

**Limitations:**

Yes.

**Quality:**

3

**Strengths And Weaknesses:**

Strengths:
- The paper is well-organized and easy to follow.
- Benchmark (3D spatial QA with multi-channel audio) is novelty.
- Training-free design is good. Leverages existing models and signal-processing methods without additional fine-tuning.
- Evaluations and ablation studies are comprehensive, which can well support the paper's main point.

Weaknesses:
- Limited scope of spatial reasoning: SAVVY-Bench only tests “which way?” and “how far?” Q&A. It doesn’t cover richer 3D reasoning tasks such as mapping unknown space, occlusion handling, trajectory prediction, semantic layout, or path planning, which limits its applicability as a general spatial reasoning benchmark.
- Pipeline Complexity: Even though it's training-free, it seems to stitch many heavy components: LLM, CLIPSeg/SAM2, depth, SLAM, SRP-PHAT, DBSCAN, Kalman. Which lead to hard to deploy and maintain.
- The paper omits inference time or resource usage, lacking assessment of real-time feasibility.
- Data is limited to indoor Aria headsets in controlled environments; outdoor generalization is unclear.

---

> ### Author Rebuttal · Authors · 2025-07-31
>
> We thank the reviewer for the helpful feedback and fruitful comments, and appreciate that the reviewer finds our paper novel in the aspect of benchmark, and notes favorably the design of the proposed training-free method. We address the questions and concerns below.
>
> **1. Limited Reasoning Scope.**
>
> We view SAVVY as a **modular, extensible foundation for future 3D spatial reasoning benchmarks**—including occlusion-aware reasoning, semantic layout, and trajectory forecasting. The evaluated tasks aim to represent realistic interplay between audio-visual-language understanding. While more abstract reasoning is valuable, our focus is on fundamental skills—such as direction, distance, temporal grounding and grounded object referral—that are critical to real-world applications and pave the way for broader reasoning capabilities. During our work on this paper, we find that even such fundamental tasks, without further complexities, require novel approaches in multi-modal learning. We preferred to focus on these fundamental tasks, proposing the first benchmark (SAVVY-Bench) and corresponding approach (SAVVY).
>
> Noteworthy, we do introduce a sufficient amount of complexity into the benchmark. **Beyond direction and localization QAs, our benchmark also introduces Snapshot Descriptor Evaluation (SD-Eval) in Table 3 (Section 4.2)**, covering: **(i) Temporal Grounding** – measurement of how accurately a model would localize queried sound events in time. **(ii) Object Referral** – tests whether the model correctly describes objects given the question and the video. We also propose a new localization accuracy metric comparing predicted and ground truth positions to test SAVVY and its variants.
>
> Indeed, most 7B open-source models achieve <5% temporal mIoU, under 35% referral accuracy, and perform only slightly above chance on direction and distance QA—highlighting that current open-source AV-LLMs still struggle with temporal alignment, semantic grounding, and basic audio spatial reasoning. These limitations make it difficult for these models to be evaluated meaningfully on more complex tasks beyond what SAVVY-Bench currently covers.
>
>
> **2. SAVVY Complexity and Runtime Efficiency.**
>
> We acknowledge that SAVVY is a multi-stage pipeline. The multiple-stage design is necessary due to the complexity of multi-modality in the task setup. However, an important advantage of the pipeline is that it is designed to be training-free and modular, which allows components to be adapted or replaced as needed. Each stage serves a targeted role in spatial reasoning (e.g., segmentation for dynamic object tracking, audio cues for out-of-view events). Furthermore, such a structured pipeline also lays the groundwork for future research on reasoning knowledge distillation as well as chain-of-thought in AV-LLMs.
>
> In regards to runtime efficiency, we include detailed latency and peak memory usage metrics below (will also be incorporated in revised version of the manuscript).
> The time and memory costs (where applicable) for each step of the SAVVY pipeline are:
> + SLAM: real-time slam on device (aria glasses).
> + Stage 1 Step 1 (Snapshot Descriptor): Runs the AV-LLM once per QA to determine temporal spans and object roles—this + matches standard AV-LLM inference latency on our benchmark (Table 2 and 3).
> + Stage1 Step 2 (Text-Guided Snapshot Segmentation with Depth Estimation): Segmentation (CLIPSeg/SAM2) is the most time-consuming (~0.52s for one frame at 512×384 per object), but we show in Supp. Table 2 that using just 32 frames uniformly sampled in one video still yields strong performance.
> Depth estimation takes ~0.44s per frame and ~6GB GPU memory; segmentation up to ~9.4GB.
> The speed and memory were tested on one NVIDIA A100 GPU, average on 200 testing samples.
> + Stage1 Step 3 (Spatial Audio Cues): Lightweight—0.06s per second of audio, CPU-only.
> + Stage 2 (Global Map Construction): Fast and efficient—0.08s per video, CPU-only.
>
> Importantly, removing the segmentation step (Table 6) significantly improves efficiency, while still outperforming LLM-only baselines, especially on the allocentric question category. This shows SAVVY can flexibly trade off between accuracy and runtime.
>
> **3. Limited to Indoor Aria Datasets.**
>
> We acknowledge that our current dataset is based on indoor scenes captured with Meta Aria glasses. This choice allows for high-quality, spatially synchronized multi-modal data—key for establishing a strong benchmark consisting of fundamental tasks. It is noteworthy to mention that SAVVY’s design is not limited to indoor environments. The pipeline operates on spatial audio and video with 6-DoF camera poses, and can generalize to outdoor or driving scenarios when higher-noise, less-structured data becomes available. We view this as an important next step in extending SAVVY-Bench toward more diverse, real-world settings.
>
> **4. Quantify the Accuracy of Zero-Shot LLM-based Event Timestamps.**
>
> **We evaluated temporal event grounding accuracy in Table 3; detailed settings are provided in lines 226–229 of the paper.** This task measures how accurately a model localizes the queried sound event in time, using IoU between predicted and ground-truth intervals. We report Recall@1, averaged over IoU thresholds from 0.05 to 0.5, summarized as mean temporal IoU (t-mIoU).
> As the reviewer points out, temporal alignment is important. Results show that most open-source 7B models achieve under 5% t-mIoU, indicating poor event-time alignment. Synchronizing complex events like speech remains a major challenge at this scale. To further investigate, we conduct experiments using the ground-truth Target Clip (i.e., the relevant event segment only), as shown in Supplementary Table 5. Results confirm that temporally aligned input consistently improves direction accuracy across all models compared to full-video input.

---

> > ### Comment · Reviewer_cjrp · 2025-08-05
> >
> > The rebuttal has resolved my primary concerns; I have raised my score to acceptance.

---

### Official Review · Reviewer_jLXv · 2025-07-05

**Clarity:** 3
**Significance:** 3
**Originality:** 4
**Rating:** 5
**Confidence:** 4

**Summary:**

This paper proposed a new dataset, named as SAVVY-Bench, for visual-audio based 3d spatial reasoning in dynamic audio-visual environments. The problem introduced by the benchmark is interesting and important for community. Besides, the authors also proposed a novel pipeline SAVVY, including egocentric spatial tracks estimation stage and dynamic global map construction for visual-audio based 3d spatial reasoning with AV-LLM. The results on SAVVY-Bench demonstrate the effectiveness of the proposed method.

**Questions:**

1）The main question I considered is the synergy between visual and audio signals. Many of the tasks presented in the paper can be solved using visual signals alone, so it is important to clarify what complementary information the audio signals provide comparing with the method using videos to build the scene and preform 3d reasoning, and how much additional value they contribute to the overall task.
2）In this benchmark, there are many 3D scene reasoning tasks. In this context, would it be possible to incorporate step-by-step reasoning similar to Chain-of-Thought approaches? Moreover, could multi-round reasoning be applied in this process? Multi-round reasoning is particularly important for embodied tasks.

**Ethical Concerns:**

["NO or VERY MINOR ethics concerns only"]

**Final Justification:**

The authors have addressed most of my concerns. As the authors proposed in rebuttal, I think the benchmark and method proposed in this work are good starting points for dynamic audio-visual environment understanding and reasoning, and will bring some inspiration to the community. Therefore, I raised my rating to Accept.

**Limitations:**

The main limitation of this paper is that it only compares reasoning performance based on large language models. To some extent, this is not entirely fair. The mapping and scene construction methods proposed in the paper could potentially be completed using only video input. Moreover, the relevant information extracted from these processes could also be incorporated into large model-based reasoning. A more detailed and comprehensive comparison is needed to better evaluate the effectiveness of the proposed approach.

**Paper Formatting Concerns:**

There is no formatting issues in this paper.

**Quality:**

3

**Strengths And Weaknesses:**

Strengths
This paper addresses a very interesting and important problem: how to perform 3D reasoning in dynamic environments with multimodal inputs, specifically audio and text. The authors provide a thorough analysis of this problem and propose a benchmark to facilitate further study. In addition, they introduce a localization method that leverages large models. The final results demonstrate the effectiveness of the proposed approach.

Weaknesses:
1）This paper mainly focuses on the audio modality. However, in real-world 3D reasoning tasks, visual signals play a crucial role. The paper lacks sufficient comparison between audio and visual signals—for example, the advantages of using audio signals versus purely visual signals, and whether combining audio and visual signals provides complementary benefits. It is also unclear whether a purely video-based approach would already be sufficient to address the challenges raised by the benchmark. More analysis is needed in this regard.
2）The reasoning tasks discussed in this paper are primarily oriented toward navigation and spatial localization. It would be interesting to explore whether audio signals can support higher-level semantic reasoning tasks, such as object grounding via sound, scene-level question answering (e.g., “How many people are there?”), or attribute reasoning (e.g., distinguishing between male and female voices).
3）The generalizability of the proposed method needs further evaluation. Since the method is training-free, it is important to investigate whether it can be applied to outdoor environments or driving scenarios.

---

> ### Author Rebuttal · Authors · 2025-07-31
>
> We thank the reviewer for providing detailed and helpful feedback, and we are glad that the reviewer thinks that our work is interesting, provides thorough analysis, and the proposed method is effective. We address the questions and concerns below.
>
> **1. Synergy between Visual and Audio Signals.**
>
> + **Limitations of Pure Visual Input:** While scene construction and 3D reasoning can, in principle, be approached using video inputs only, relying solely on visual signals typically introduces limitations—especially in the context of audio-centric questions and events. **Firstly, questions from SAVVY-Bench focus on audio events** such as speech, making them difficult to answer using visual input alone. **Secondly, visual cues are often unreliable in out-of-view or occluded scenarios.** When the sound source disappears from view for extended periods, the model inaccurately estimates its trajectory based on its last visible location.
>
> + **Blind Testing (Pure Audio Input):** As shown in Supplementary Table 4, the blind testing that we conducted reveals that adding visual input to pure audio input can offer limited benefit for 7B open-source models, with increased benefit for models such as Gemini 2.5 Pro, which possess stronger visual understanding and temporal reasoning capabilities. For example, in Section 5.4, we show that our testing reveals that Gemini-2.5-Pro significantly relies on visual input to localize sound sources—linking audio to visible objects, grounding event time, and inferring direction.
>
> + **Complementary information from combining visual and audio:** **Spatial audio naturally captures off-screen and out-of-view information, providing continuity where visual input cannot provide it. SAVVY-Bench highlights such challenging though typical cases and SAVVY demonstrates that integrating spatial audio cues with vision enables robust reasoning in these situations.**
> While human visual attention and egocentric cameras (e.g., Aria’s 110° HFOV) are restricted to a limited frontal field of view, the acoustic environment inherently provides 360-degree coverage. Auditory cues—such as appliance alerts, speech from adjacent rooms, or footsteps from behind—convey critical contextual information beyond the visible scene, highlighting the role of spatial audio in achieving a more complete perception of real-world environments.
> This is key for developing safer and more reliable spatial intelligence in multi-modal LLMs. Our results underscore the complementary strengths of both modalities and point to promising directions for future research in multi-modal spatial reasoning.
>
> **2. Limited Reasoning Tasks.**
>
> The evaluated tasks aim to represent realistic interplay between audio-visual-language understanding. While more abstract reasoning (e.g., voice gender classification or crowd counting) is valuable, **our focus is on fundamental skills—such as direction, distance, temporal grounding, and grounded object referral—that are critical to real-world applications and pave the way for broader reasoning capabilities.** During our work on this paper we found that even such fundamental tasks, without further complexities, require novel approaches in multi-modal learning and preferred to focus on these as a first benchmark (SAVVY-Bench) and approach (SAVVY).
>
>
> Noteworthy, we do introduce some amount of complexity into the benchmark. **Beyond direction and localization QAs, our benchmark also introduces Snapshot Descriptor Evaluation (SD-Eval) in Table 3 (Section 4.2)**, covering: **(i) Temporal Grounding** – measurement of how accurately a model would localize queried sound events in time. **(ii) Object Referral** – tests whether the model correctly describes objects given the question and the video. We also propose a new localization accuracy metric comparing predicted and ground truth positions to test SAVVY and its variants.
>
> Indeed, most 7B open-source models achieve <5% temporal mIoU, under 35% referral accuracy, and perform only slightly above chance on direction and distance QA—highlighting that current AV-LLMs still struggle with temporal alignment, semantic grounding, and basic audio spatial reasoning. These limitations make it difficult to meaningfully evaluate more complex tasks beyond what SAVVY-Bench currently covers.
>
> **3. Generalizability to Diverse Scenarios.**
>
> While our current dataset focuses on high-quality indoor scenarios, the SAVVY model and input/output formats are not inherently limited to indoor environments. When evaluating LLMs, we use stereo audio and rectified camera views to promote broader applicability.
>
> We acknowledge that full generalization to outdoor or driving environments is challenging, primarily due to degraded audio quality in such settings (e.g., wind, road noise). Our current focus is on high-quality, controlled indoor scenarios as a foundational step for studying spatial audio-visual reasoning. This allowed us to establish a reliable benchmark and isolate core model capabilities. That said, we see this as the starting point of an iterative process.
>
>
> **4. Step-by-Step Reasoning.**
>
> We thank the reviewer for this proposal and fully agree—supporting step-by-step and multi-round reasoning is one of the key motivations behind the SAVVY pipeline and SAVVY-Bench. While current models have plenty of room for improvement with grounding and spatial alignment, **we see a great potential in future work that distills SAVVY-style reasoning chains into LLMs to enhance their spatial and temporal understanding with audio and visual modality**. Our benchmark provides a structured foundation to study such reasoning processes, and we hope it encourages future research in this direction—especially toward more embodied and interactive tasks.
>
> Specifically, our pipeline naturally decomposes spatial reasoning into interpretable steps:
> + Temporal grounding (when did the event occur?)
> + Object referral (what objects are involved?)
> + Spatial localization (where are they in 3D space?)
> + Trajectory tracking (how do they move over time?)
> + Spatial relationship reasoning (what are the relative distance and direction?)
>
> Each intermediate output in SAVVY represents a reasoning step that current end-to-end models struggle to produce explicitly. These outputs could be used to create training data for teaching LLMs to perform structured spatial reasoning.

---

> > ### Comment · Reviewer_jLXv · 2025-08-08
> >
> > The authors have addressed most of my concerns，and thus I will increase my rating to accept

---

### Note · Authors · 2025-08-15

We sincerely thank all reviewers for their thoughtful, constructive, and positive feedback on our submission. We are grateful that our work has been recognized as both novel and impactful, and we appreciate the acknowledgment of our rigorous benchmark construction, thorough analysis, and principled pipeline design.

**Our Key Contributions:**
+ **First-of-its-kind Benchmark**: SAVVY-Bench is the first benchmark to rigorously evaluate 3D spatial reasoning in dynamic, real-world audio-visual environments, with synchronized spatial audio and fine-grained temporal grounding. This fills a critical gap in multi-modal LLM evaluation and sets a new standard for future research.
+ **Training-Free, Modular Reasoning Pipeline**: SAVVY introduces a novel, training-free pipeline that integrates state-of-the-art AV-LLMs with structured spatial reasoning, leveraging both visual and audio cues. Our approach demonstrates substantial improvements over existing baselines and provides a flexible, extensible foundation for future advances in multi-modal spatial intelligence, including enhanced reasoning capabilities.
+ **Comprehensive Analysis and Ablations**: We provide detailed quantitative and qualitative evaluations, including ablation studies and various metrics, to support the robustness and practical relevance of our approach. Our results highlight the complementary strengths of audio and visual modalities, and reveal key bottlenecks in current AV-LLMs.
+ **Catalyzing Future Research**:  By formalizing new tasks and reasoning structures, our work opens up promising directions for chain-of-thought reasoning, knowledge distillation, and real-world embodied AI applications, with a particular focus on advancing spatial reasoning in audio-visual modalities.

We sincerely appreciate the reviewers’ insightful suggestions, which have helped us further clarify our contributions and strengthen our manuscript. We believe we have addressed major concerns, including efficiency, generalizability, and the value of multi-modal integration.
We are confident that SAVVY-Bench and SAVVY will serve as a foundational contribution to the next generation of audio-visual spatial reasoning research, and we look forward to seeing the community build upon these results.


Thank you for your consideration.

---

### Decision · Program_Chairs · 2025-09-17

**Decision:**

Accept (oral)

**Comment:**

(a) The authors' claims include -- spatialization task via localization cues as a candidate for inducing reasoning in LLMs, using perception from localization as a way to training AV models, connecting training free optimization in LLMs to spatial audio tasks (first of their kind), novel AV model that can handle dynamics in scene via prompting. All these are well validated claims and the paper's novelty and evaluations justify the claims.

(b) There are a number of important things the work is highlighting
- spatialization is implicit in LLMs if we can probe/create a prompt that is 'detailed' enough abut context
- introducing a new benchmark for 3d spatial reasoning in dynamics -- this is a very hard problem, relatively less is known on it
- training-free methods and DSP/perception driven tuning helps scale and adopt the work into other areas in future
- generally rigorous evaluations

(c) Given the new-ness of this work / area, some of these are not really weakness, rather potential directions that the field/others can take in future : extension to more complicated tasks beyond simple navigation and localization tasks (e.g., what happens next, can you chart a path for me to take etc.); generalization of the proposal model for large-scale training of AV-LLMs i.e., connecting the dots to training driven reasoning models from this training-free setup; directly inducing physics information into the optimization constraints for DPO (again moving into training-dependent setups)

(d) The authors tackle a difficult reasoning problem (spatialization in context of LLMs), bridge human perception style thinking into machine perception (navigation driven tasks and localization cues into prompting), build a training free method (scales well, modular) and rigorously evaluate the proposal; and the paper is written well. This work will influence future works on perception driven reasoning in language models. It ticks all the bars for an oral.

(e) All reviewers agree on the strengths and impact of the work. Most of the discussion was about clarification of evaluations, modeling choice, simplicity of task definition, generalization to other tasks etc. These were addressed.